# REDf: a deep learning model for short-term load forecasting to facilitate renewable integration and attaining the SDGs 7, 9, and 13

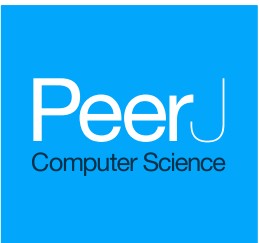

Md Saef Ullah Miah[1], Junaida Sulaiman[2], Md Imamul Islam[3], Md Masuduzzaman[4], Molla Shahadat Hossain Lipu[5] and Ramdhan Nugraha[6]

[1] Department of Computer Science, American International University—Bangladesh, Dhaka, Dhaka, Bangladesh
[2] Faculty of Computing, Universiti Malaysia Pahang Al-Sultan Abdullah, Pekan, Pahang, Malaysia
[3] Faculty of Electrical and Electronics Engineering Technology, Universiti Malaysia Pahang Al-Sultan Abdullah, Pekan, Pahang, Malaysia
[4] Department of Degroote School of Business, McMaster University, Hamilton, Canada
[5] Department of Electrical and Electronic Engineering, Green University of Bangladesh, Dhaka, Dhaka, Bangladesh
[6] Faculty of Electrical Engineering, Telkom University, Bandung, Indonesia

Corresponding authors
Md Saef Ullah Miah, saef@aiub.edu
Ramdhan Nugraha, ramdhan@telkomuniversity.ac.id

## ABSTRACT

Integrating renewable energy sources into the power grid is becoming increasingly important as the world moves towards a more sustainable energy future in line with the United Nations (UN) Sustainable Development Goal (SDG) 7 (Affordable and Clean Energy). However, the intermittent nature of renewable energy sources can make it challenging to manage the power grid and ensure a stable supply of electricity, which is crucial for achieving SDG 9 (Industry, Innovation and Infrastructure). In this article, we propose a deep learning model for predicting energy demand in a smart power grid, which can improve the integration of renewable energy sources by providing accurate predictions of energy demand. Our approach aligns with SDG 13 (Climate Action) on climate action, enabling more efficient management of renewable energy resources. We use long short-term memory networks, well-suited for time series data, to capture complex patterns and dependencies in energy demand data. The proposed approach is evaluated using four historical short-term energy demand data datasets from different energy distribution companies, including American Electric Power, Commonwealth Edison, Dayton Power and Light, and Pennsylvania-New Jersey-Maryland Interconnection. The proposed model is compared with three other state-of-the-art forecasting algorithms: Facebook Prophet, support vector regression, and random forest regression. The experimental results show that the proposed REDf model can accurately predict energy demand with a mean absolute error of 1.4%, indicating its potential to enhance the stability and efficiency of the power grid and contribute to achieving SDGs 7, 9, and 13. The proposed model also has the potential to manage the integration of renewable energy sources effectively.

## INTRODUCTION

The world is currently experiencing a serious dilemma in the energy field due to the rapid depletion of fossil fuels due to rising populations, urbanization, and technological advancements. In addition, burning fossil fuels results in water and air contamination, climate change, and the production of greenhouse gases, all of which contribute to the acceleration of global warming and have severe adverse effects on ecosystems and human health (*Islam et al., 2023b*). To mitigate the impacts of climate change, scientists, academics, and policymakers are working to mainstream renewable energy (RE) as a replacement for carbon-based power sources (*Islam et al., 2023a*). To achieve the 1.5 °C scenario goal by 2050, the most significant threshold is to ensure 90% electricity generation from RE sources and 79% of the overall energy consumption (*IRENA, 2022*). Over the last decade, the installation and generation of renewable energy in terms of off-grid and on-grid systems have increased significantly. According to the International Renewable Energy Agency (IRENA), the latest trends in renewable energy are shown in Fig. 1 (*IRE Agency, 2023*). This figure shows that the maximum number of installations get direct connections to the grid, and the increase in solar and wind-based power plants is noticeable compared to other technologies. Portions of this text were previously published as part of a preprint (https://arxiv.org/pdf/2304.03997v3).

### Background

The imperative transition towards a sustainable future necessitates integrating renewable energy sources into power grids, reshaping global energy systems (*Al-Shetwi, 2022*; *Jenkins, Long & Wu, 2015*). Aligned with the United Nations' (UN) ambitious goal of universal access to affordable, reliable, and modern energy by 2030 (*United Nations, 2023*; *He et al., 2022*), this integration emerges as pivotal for sustainable development. The UN Sustainable Development Goals (SDGs) set the framework, with SDG 7 (Affordable and Clean Energy), SDG 9 (Industry, Innovation and Infrastructure), and SDG 13 (Climate Action) standing as critical benchmarks for an equitable and environmentally sustainable energy landscape (*Busch & Wydra, 2023*).

Deep learning models play a crucial role in achieving these SDGs, particularly in predicting energy demand with precision. This accuracy enables optimal management of renewable resources, diminishes reliance on fossil fuels, reduces greenhouse gas emissions, and fortifies resilient and efficient infrastructure. Achieving high shares of renewable energy, as emphasized by SDG 7, diversifies the energy mix, decreases costs, and enhances accessibility.

Extending the focus to SDG 9, the study underscores the importance of fostering sustainable infrastructure and promoting innovation in the energy sector. Additionally, SDG 13 becomes a compelling imperative as the transition to renewable energy significantly contributes to mitigating the impacts of climate change.

Despite the promise of renewable sources, challenges persist, notably the unpredictability of energy generation (*Ali et al., 2021*; *Maier & Gemenetzi, 2014*). Addressing this challenge requires accurate predictions of energy demand, aligning with

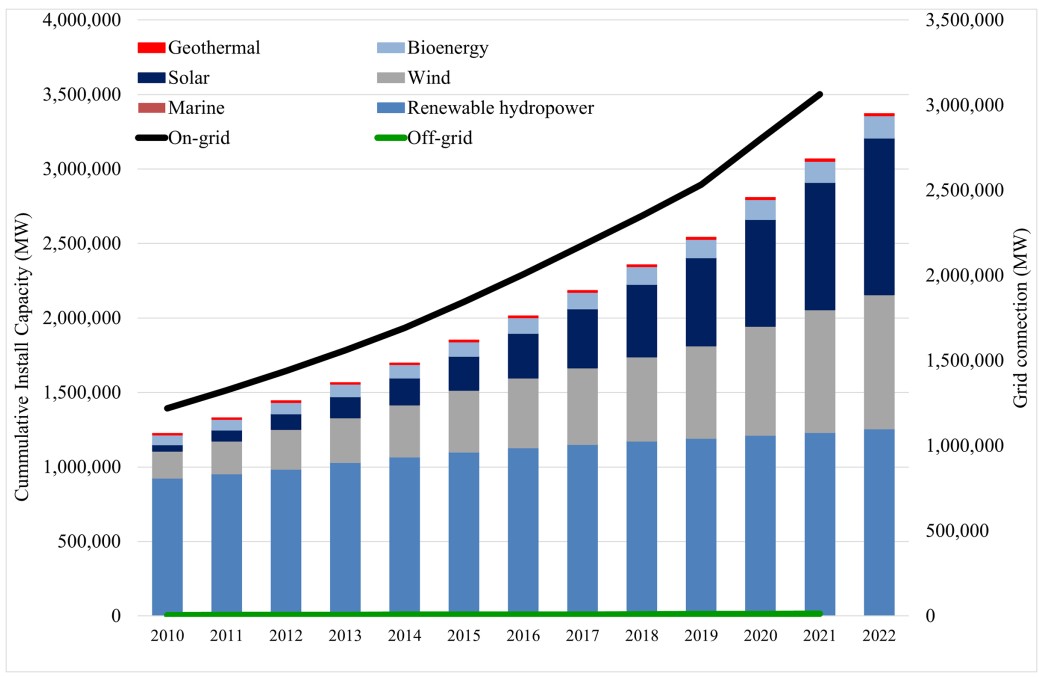

**Figure 1 Global renewable energy trend and grid application scenario.**

SDG 7. Developing effective predictive models becomes paramount in realizing these objectives.

This study delves into the intricate landscape of facilitating greater integration of renewable energy by predicting aggregated energy demand, aligning with multiple SDGs crucial for constructing an equitable and sustainable energy future (*Aydos et al., 2022*).

Harnessing clean energy sources such as hydroelectricity, geothermal, biomass, solar, and wind power not only reduces energy prices but also assures a dependable and sustainable energy supply globally (*Perera, Coccolo & Scartezzini, 2019*; *United Nations Department of Economic and Social Affairs, 2017*). Addressing climate change through the grid integration of renewable energy becomes pivotal for reducing greenhouse gas emissions (*Giri et al., 2022*) and mitigating its adverse effects (*Joseph & Inambao, 2020*). The transition from fossil fuels to renewable energy stands as a tangible step in fulfilling climate commitments under the Paris Agreement, safeguarding the planet for future generations (*Akaev & Davydova, 2020*).

While the integration of renewable sources holds promise, challenges persist, notably the unpredictability of energy generation (*Ali et al., 2021*; *Maier & Gemenetzi, 2014*). Tackling this challenge necessitates predicting energy demand using deep learning for a smart power grid, reflecting SDG 7. The development of accurate predictive models is a critical enabler in achieving this.

## Literature review

The rapid advancement of machine learning techniques has significantly transformed the energy sector, offering innovative solutions to complex challenges such as energy

consumption prediction, load forecasting, and efficient energy management. As the demand for sustainable and efficient energy systems grows, researchers have increasingly turned to advanced computational models, including deep learning and hybrid approaches, to enhance the accuracy and reliability of energy-related predictions. This literature review explores recent developments in the application of machine learning techniques to energy systems, categorizing the reviewed studies based on their methodologies, applications, and contributions. From hybrid deep learning models for household energy consumption to long short-term memory (LSTM)-based frameworks for smart grid applications, the reviewed works highlight the transformative potential of machine learning in addressing critical energy challenges. The following subsections provide a detailed analysis of these studies, organized by their thematic focus and methodological innovations.

### Hybrid deep learning models for energy consumption prediction

Machine learning techniques have garnered considerable attention in the energy sector recently. Within this evolving landscape, several noteworthy innovations have come to the forefront. For instance, an innovative hybrid deep learning model, which combines LSTM (*You et al., 2023*) neural networks with the stationary wavelet transform, has demonstrated its ability to enhance predictions of household energy consumption. This model addresses challenges associated with irregular behaviors and univariate data issues (*Yan et al., 2019*). Additionally, in the context of developing countries, the application of artificial neural networks (ANNs) (*Giri et al., 2023*) coupled with meta-heuristic techniques has proven effective for forecasting load demand. This approach is pivotal in guiding energy-efficient growth in regions where technical resources and infrastructure are limited (*Arnob et al., 2023*). Furthermore, a novel framework has been introduced to streamline residential energy management. This framework integrates an ANN-based forecast engine with a controller using the Differential Evolution Algorithm modified for Grey Wolves (DA-GmEDE). The outcome is a substantial reduction in energy costs and peak-to-average ratios, emphasizing the potential for cost-effective and efficient energy management within the smart grid (*Hafeez et al., 2020*).

### Deep learning models for smart grid applications

The authors of the article (*Amalou, Mouhni & Abdali, 2022*) analyzed the efficiency of different deep learning models, including recurrent neural network (RNN), LSTM, and gated recurrent unit (GRU), in predicting energy demand for the Smart Grid Smart City project using datasets from 2010 to 2014. The models are evaluated using root mean squared error (RMSE), mean absolute error (MAE), and coefficient of determination ($R^2$) scores. The results showed that GRU outperforms basic RNN and LSTM with the lowest RMSE and highest $R^2$ score due to its ability to deal with the vanishing gradient problem and its impact on the number of parameters.

In another work (*Alrasheedi & Almalaq, 2022*), the authors proposed using hybrid deep learning methods to improve load forecasting accuracy in the Saudi smart grid system. It aims to develop reliable forecasting models and understand the relationship between

various features and attributes in the Saudi smart grid. The model uses a real dataset from Jeddah and Medinah grids for an entire year with a one-hour time resolution and compares prediction results with conventional deep learning methods, including RNN (*Hopfield, 1982*), LSTM (*Hochreiter & Schmidhuber, 1997*), GRU (*Dey & Salem, 2017*), and convolutional neural network (*LeCun et al., 1998*). The results show that the proposed hybrid deep learning models, particularly CNN-GRU and CNN-RNN, provide 1.4673% and 1.222% improvement in load forecasting accuracy, respectively, compared to the benchmark strategy.

### LSTM-based models for household energy consumption

*Shachee, Latha & Hegde Veena (2022)* used LSTM-RNN-based deep learning architecture to predict household electrical energy consumption two months in advance. The model is trained on relevant features and evaluated by comparing actual and predicted values. The proposed model helps households conserve energy and is evaluated using the UCI repository dataset of domestic electric consumption (*Dua & Karra Taniskidou, 2017*). The results showed that the LSTM model has much higher precision than statistical and engineering prediction models, with a compatible RMSE of 0.6 compared to conventional models.

The study by *Taleb et al. (2022)* proposed a hybrid machine learning model that combines standard neural networks with an automatic weight update process based on past errors. This flexible model can predict energy demand over various time ranges and regions. The effectiveness of the proposed model was demonstrated by achieving a MAE of 372.08 in energy demand prediction for Mayotte Island.

### Residential load forecasting with LSTM models

Residential load forecasting is becoming increasingly important as smart meters are increasingly deployed at the household level to collect historical data on energy consumption. In the study by *Mubashar et al. (2022)*, they proposed a method for load forecasting and validated it using real-world data sets. They compared the performance of their proposed method, which uses LSTM models, with two commonly used techniques, autoregressive integrated moving average (ARIMA) and exponential smoothing. They evaluated the accuracy of load forecasts generated using these three techniques using real data from 12 houses over three months. The results indicated that LSTM models performed better than the other two methods for time series-based predictions. Their model achieved an MAE of 2.44736176.

### Multivariate prediction of energy time series

Another work by *Rosato et al. (2019)* presented a novel deep learning approach for multivariate prediction of energy time series. The proposed approach utilized convolutional neural network and long short-term memory models to combine and filter several correlated time series while considering their long-term dependencies. The learning scheme is implemented as a stacked deep neural network, with one or more layers feeding their output into the input of the subsequent layer. The effectiveness and accuracy of the proposed approach are demonstrated through real-world applications in the energy sector,

highlighting its robustness and accuracy. The lowest RMSE the method achieved among all the variations tested is 2.252, achieved on a baseline 1-day forecast.

### Electricity demand forecasting with LSTM

Electric power load demand forecasting is critical for energy management, requiring accurate planning and infrastructure investment predictions. Despite a lot of research in this area, accuracy remains an issue. *Nguyen, Duong & Le (2020)* proposed an electricity demand forecasting method based on the LSTM deep learning model, tested using 6 years of power consumption data in Vietnam. The proposed method achieved an RMSE of 9.63, indicating its potential as a valuable tool for energy sector studies.

### Short-term load forecasting with advanced models

*Pramono et al. (2019)* proposed a method for short-term load forecasting using a wavenet-based model that employs dilated causal residual CNN and LSTM layers. The proposed model outperforms other deep learning-based models in terms of RMSE and MAE, achieving RMSE and MAE equal to 203.23, and 142.23 for the ENTSO-E testing dataset 1, and 292.07 and 196.95 for ENTSO-E dataset 2. For the ISO-NE dataset, the RMSE, and MAE are equal to 85.12, 58.96 for ISO-NE testing dataset 1 and 85.31, and 62.23 for ISO-NE dataset 2. The proposed method aimed to support the demand response program in hybrid energy systems, especially those using renewable and fossil sources. Two different ways of conducting model testing were conducted: one using datasets with identical distributions as the validation data and the other with unknown distributions.

### Non-intrusive attention-augmented deep learning models

This study presents a non-intrusive attention-augmented deep learning model, referred to as NAP-BiLSTM, for forecasting short-to-mid-term electricity consumption (*Li et al., 2022*). The architecture integrates a non-intrusive attention-based preprocessing (NAP) module with a bidirectional long short-term memory (BiLSTM) network. The NAP module acts as an independent attention mechanism applicable to time series analysis without altering the fundamental structure of the neural network. The model's performance is validated through two experimental setups: univariate forecasting using United States electricity consumption data and multivariate forecasting incorporating meteorological and energy data from Valencia. Experimental results indicate that NAP-BiLSTM surpasses existing deep learning models in terms of RMSE, MAE, and mean absolute percentage error (MAPE) across both prediction tasks. Furthermore, the proposed approach achieves a MAE reduction of 2.36% to 16.52% when compared to conventional models such as AM-BiLSTM, Stacked-BiLSTM, Adaboost, and attention-CNN-LSTM for mid-term multidimensional time series forecasting.

### Advantages of LSTM networks for energy demand forecasting

Compared to support vector regression (SVR) and random forest regression (RFR), LSTM networks offer several advantages for energy demand forecasting. Unlike traditional models that rely on short-term statistical patterns, LSTMs capture long-term dependencies, making them highly effective for time series forecasting. Their gating

mechanisms, particularly the forget gate, allow the model to selectively retain useful information while discarding irrelevant data, preventing challenges like the vanishing gradient problem. Additionally, energy demand patterns are highly non-linear, and LSTMs adapt dynamically through hidden states, unlike linear regression models. Furthermore, LSTMs naturally learn seasonal and cyclic behaviors, making them well-suited for applications requiring dynamic adaptation. These capabilities enable LSTM models to outperform traditional regression-based forecasting techniques, making them particularly valuable for smart grid applications.

The performance of different models studied in different works is summarized in Table 1. This table shows the comparison of the studies with respect to different evaluation metrics. From the table, it can be observed that the LSTM-RNN (*Shachee, Latha & Hegde Veena, 2022*) model did not report the MAE and $R^2$ values mentioning only RMSE. The CNN-GRU (*Alrasheedi & Almalaq, 2022*) model achieved a high $R^2$ score but did not report the MAE and RMSE values. On the other hand, the LSTM and GRU models reported MAE and $R^2$ values, but their performance was inferior to the proposed model. Apart from these, most studies did not report $R^2$ values.

The related works in this field demonstrate various approaches for predicting energy demand using machine learning techniques such as neural networks and time series analysis. These studies have shown promising results in improving energy demand forecasts, highlighting the potential of machine learning in the energy sector. However, there is still room for improvement, and further research is needed to refine and optimize these techniques to provide more accurate and reliable predictions.

## Problem statement

Machine learning techniques have gained prominence in the energy sector, showcasing innovations in predicting energy demand. Noteworthy models, such as a hybrid approach combining LSTM networks with stationary wavelet transform, address irregular behaviors but leave gaps in providing a standardized evaluation framework (*Yan et al., 2019*). Similarly, the effectiveness of ANNs in developing countries lacks comprehensive evaluation metrics, leaving uncertainties in their general applicability for load demand forecasting (*Arnob et al., 2023*; *Giri et al., 2023*).

Novel frameworks integrating ANN-based forecast engines and controllers exhibit potential for cost-effective energy management, but challenges persist in refining forecasting models and understanding feature relationships within smart grids (*Hafeez et al., 2020*). The comparative analysis by *Amalou, Mouhni & Abdali (2022)* reveals the superiority of GRU over RNN and LSTM, yet achieving accuracy in load forecasts remains a concern. The proposed hybrid deep learning models in the Saudi smart grid system show promise, but the need for further refinement to enhance forecasting accuracy is evident (*Alrasheedi & Almalaq, 2022*).

Studies predicting household energy consumption using LSTM-RNN architectures indicate higher precision but lack comprehensive reporting on metrics such as MAE and $R^2$ values, posing challenges in evaluating overall model performance (*Shachee, Latha &*

**Table 1 Performance comparison of different studies.**

| Reference | Year | Model | MAE | $R^2$ | RMSE |
| --- | --- | --- | --- | --- | --- |
| *Li et al. (2022)* | 2022 | NAP-BiLSTM | 0.02 | – | 0.027 |
| *Amalou, Mouhni & Abdali (2022)* | 2022 | LSTM | 0.021 | 0.53 | 0.039 |
| *Amalou, Mouhni & Abdali (2022)* | 2022 | GRU | 0.022 | 0.64 | 0.034 |
| *Alrasheedi & Almalaq (2022)* | 2022 | CNN-GRU | – | 0.973 | 0.816 |
| *Shachee, Latha & Hegde Veena (2022)* | 2022 | LSTM-RNN | – | – | 0.6 |
| *Taleb et al. (2022)* | 2022 | CNN-LSTM-MLP | 372.08 | – | – |
| *Mubashar et al. (2022)* | 2022 | LSTM | 2.447 | – | – |
| *Rosato et al. (2019)* | 2019 | CNN-LSTM | – | – | 2.252 |
| *Nguyen, Duong & Le (2020)* | 2020 | LSTM | – | – | 9.63 |
| *Pramono et al. (2019)* | 2019 | CNN-LSTM-ENTSO-E | 142.23 | – | 203.23 |
| *Pramono et al. (2019)* | 2019 | CNN-LSTM-ISO-NE | 58.96 | – | 85.12 |

*Hegde Veena, 2022*). *Taleb et al.*'s *(2022)* flexible energy demand prediction model shows effectiveness but raises questions about its adaptability across various contexts.

Residential load forecasting studies highlight the superior performance of LSTM models compared to traditional methods, yet discrepancies in reported metrics raise concerns about the robustness of these findings (*Mubashar et al., 2022*). Additionally, the multivariate prediction approach using CNN-LSTM models for energy time series lacks standardized benchmarks, hindering a comprehensive understanding of its effectiveness (*Rosato et al., 2019*).

Despite advancements, the accuracy and reliability of energy demand predictions remain inconsistent across various models, revealing research gaps in standardization of evaluation metrics and the need for further model refinement. The absence of a unified framework hampers comparability and generalizability of results, necessitating comprehensive research to address these gaps and enhance the overall efficiency and sustainability of energy systems.

## Key contributions

In alignment with recent studies and the defined objectives, this research introduces a novel approach utilizing deep learning, a sub-field of machine learning tailored for analyzing sequential data, particularly time series data (*Ismail Fawaz et al., 2019*; *Hossain et al., 2022*). The proposed methodology leverages a LSTM network, a type of RNN explicitly designed to model sequential data with temporal dependencies (*Hochreiter & Schmidhuber, 1997*). Training the model involves utilizing historical energy demand data, and its performance is rigorously evaluated using metrics such as MAE, RMSE, and $R^2$.

In addition to its prowess in accurately predicting energy demand, our proposed method showcases robust generalization capabilities for previously unobserved data. The ability to forecast energy demand accurately can revolutionize power grid management, fostering more efficient distribution and utilization of renewable energy sources while diminishing reliance on nonrenewable alternatives. Moreover, the proposed method has

the capacity to enhance overall power infrastructure efficiency, cut costs, and facilitate the seamless integration of renewable energy sources.

The key contributions of this work extend beyond technical advancements to encompass a positive impact on SDGs, particularly SDG 7 (Affordable and Clean Energy), SDG 9 (Industry, Innovation, and Infrastructure), and SDG 13 (Climate Action). These contributions align with the United Nations' broader agenda for sustainable development. The specific contributions are as follows:

- **Introduction of LSTM-based deep learning model:** The proposed deep learning model addresses the pivotal SDG 7, ensuring affordable and clean energy for all. By enhancing the precision of energy demand forecasts, the model promotes efficient utilization of renewable energy sources, thus contributing to a sustainable and accessible energy future.
- **Minimization of MAE:** The emphasis on minimizing MAE in the energy demand prediction model directly supports SDG 9 by fostering innovation and infrastructure development. Accurate energy forecasts enable infrastructure optimization, paving the way for more sustainable and resilient energy systems.
- **Comprehensive model evaluation:** The rigorous evaluation of the proposed approach using various metrics, including MAE, RMSE, $R^2$, and MAPE aligns with SDG 13. By improving the accuracy of energy demand predictions, the model contributes to climate action efforts, reducing reliance on nonrenewable energy sources and mitigating environmental impacts.

In essence, this article's primary contributions lie in technical advancements and in addressing global challenges outlined in the SDGs. The research aligns with the United Nations' vision for a sustainable future, emphasizing the role of accurate energy demand forecasting in achieving key sustainable development objectives.

The remaining sections of this article are structured as follows. The study's materials and methods, including data acquisition and pre-processing, model development, evaluation, and deployment, are described in detail in the following section. The experimental findings are then presented in the results and discussion section, along with a comparison to other existing methods. Finally, the study concludes with recommendations for future research.

## MATERIALS AND METHODS

In this study, we propose an approach based on deep learning for forecasting energy demand in a smart grid. The primary objective of this strategy is to improve the integration of renewable energy sources by providing accurate energy demand forecasts that can aid in the administration of the power grid. The proposed method continues with model evaluation and deployment, beginning with data collection, pre-processing, and model development. Starting with the formulation of the problem, we will describe each phase in detail and explain the tools and techniques used at each stage. The overview of the methodology employed in this study is shown in Fig. 2.
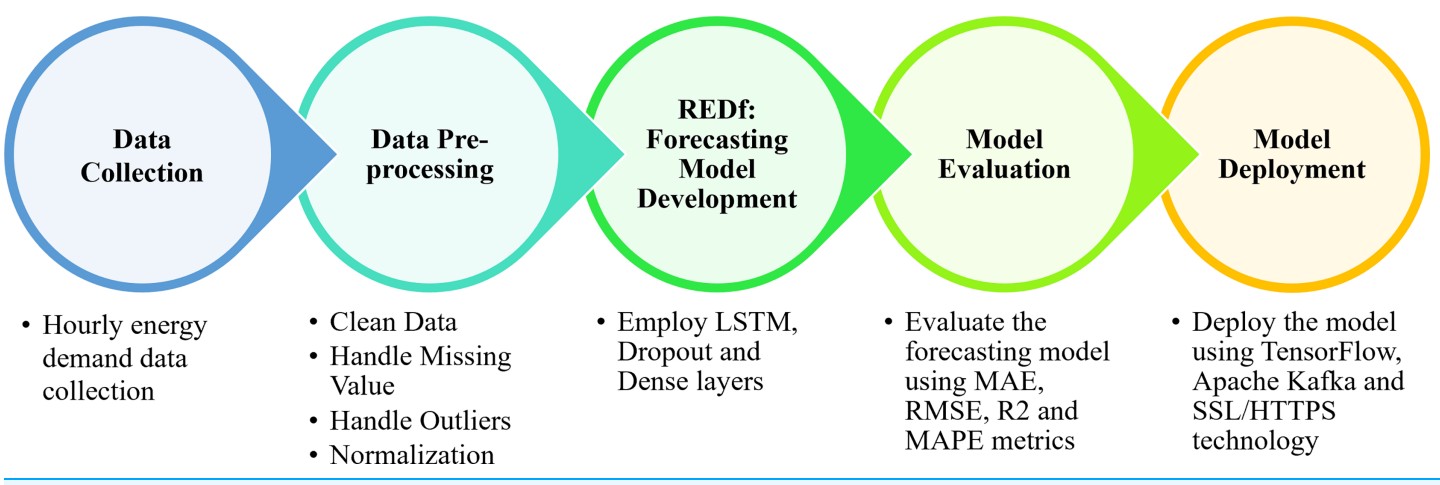

**Figure 2 Overview of the methodology employed in this study.**

## Problem formulation

The problem formulation section provides the foundation and context for the proposed solution in the study by clearly defining the objective, scope, and challenges of the energy demand forecasting problem. The problem of predicting energy demand for a smart power grid can be formulated mathematically as follows:

Given a time series of historical energy demand data, represented by a sequence of vectors $<D = d_1, d_2, \ldots, d_n>$, where $d_i$ is a vector of energy demand values at time $i$, the goal is to predict the energy demand at a future time, represented by a vector $d_{n+h}$, where $h$ is the number of time steps ahead for which the prediction is made.

This problem can be represented as a function $f(D) = d_{n+h}$, where the function $f$ maps the historical energy demand data to the predicted energy demand.

The objective is to find the optimal function $f$ that minimizes the prediction error, which can be defined as the MAE between the predicted and actual energy demand values.

## Data collection

Data collection is essential in predicting energy demand in a smart power grid. This section briefly describes the steps and techniques used for data collection in the proposed study.

The initial phase of data collection involves the identification of pertinent data sources, necessitating the recognition of data types essential for enabling precise predictive modeling, including variables such as energy consumption, meteorological conditions, and economic indicators. Data acquisition encompasses a diverse array of potential sources, ranging from utility providers and government agencies to publicly accessible datasets. In the context of this research, the study leverages hourly energy demand data procured from American Electric Power (AEP), a prominent electric utility enterprise within the United States, renowned for its extensive service coverage encompassing over 5 million customers across 11 states (*AEP, 2023*). The dataset in question encompasses a robust repository comprising 121,273 data entries, documenting hourly energy consumption spanning from

December 2004 to January 2018. Furthermore, to assess the model's performance comprehensively, the supplementary datasets COMED (https://www.comed.com/Pages/default.aspx), DAYTON (https://www.aes-ohio.com/about-aes-ohio), and PJME (https://pjm.com/) have been enlisted for evaluation. These datasets collectively serve as established benchmarks for scrutinizing the efficacy of energy demand forecasting models and are readily accessible through the following GitHub repository: https://github.com/panambY/Hourly_Energy_Consumption (*Pereira, 2023*).

COMED hourly energy consumption data refers to the hourly electricity consumption data for the Commonwealth Edison (COMED) service area, which covers the northern part of Illinois in the United States. The data provides information on the hourly electricity demand for residential, commercial, and industrial customers in the COMED service area. This dataset contains historical energy consumption data in an hourly fashion from December 2011 to January 2018, with a total of 66,497 data points.

DAYTON hourly energy consumption data refers to the hourly electricity consumption data for the Dayton Power and Light (DP&L) service area, which covers the city of Dayton, Ohio, and surrounding areas in the United States. The data provides information on the hourly electricity demand for residential, commercial, and industrial customers in the DP&L service area. This dataset contains historical energy consumption data in an hourly fashion from December 2004 to January 2018, with a total of 121,275 data points.

PJME hourly energy consumption data refers to the hourly electricity consumption data for the Pennsylvania-New Jersey-Maryland Interconnection (PJM) regional transmission organization in the United States. The data provides information on the hourly electricity demand for a large portion of the eastern United States, covering 13 states and the District of Columbia, and includes variables such as date, time, temperature, and electricity demand. This dataset contains historical energy consumption data in an hourly fashion from December 2002 to January 2018, with a total of 145,366 data points. Table 2 shows an overview of the datasets utilized to benchmark the proposed model.

After data has been collected, it has to be pre-processed to ensure that it is in a format compatible with the model. This includes removing any inconsistencies or errors from the data and transforming the data into a format that the model can use. The data pre-processing procedures described in the following section describe normalisation, feature scaling, and outlier removal.

## Data pre-processing

Data pre-processing is essential in predicting energy demand in a smart power grid. It ensures that the data used to train and evaluate the model is clean, consistent, and in a suitable format for the model. This section will describe the steps and techniques used for data pre-processing in the proposed study.

The initial phase in data pre-processing is data cleansing. This includes eliminating any data inconsistencies, errors, or missing values. Common data cleansing techniques include removing duplicates, replacing absent values with imputed values, and converting data to a standard format. In our case, for all the datasets employed in this study do not contain any

**Table 2 Overview of the datasets utilized in this study.**

| Dataset | Start datetime | End datetime | Data points |
|---------|----------------|--------------|-------------|
| AEP | 2004-12-31 01:00:00 | 2018-01-02 00:00:00 | 121,273 |
| COMED | 2011-12-31 01:00:00 | 2018-01-02 00:00:00 | 66,497 |
| DAYTON | 2004-12-31 01:00:00 | 2018-01-02 00:00:00 | 121,275 |
| PJME | 2002-12-31 01:00:00 | 2018-01-02 00:00:00 | 145,366 |

missing values and any outlier. To check and detect the missing values and outliers "isnull" and "IsolationForest" techniques are employed from "Scikit-learn" library.

The following phase transforms the data into a format the model can utilize. This includes the normalization, scaling, and encoding of categorical variables. Normalization is adjusting the data so that the mean is 0 and the standard deviations are 1. Scaling the data can prevent the magnitude of the data from affecting the model. In this study, the "sklearn.preprocessing.MinMaxScaler" is employed to normalize the data. Figure 3 illustrates an example using the PJME dataset, comparing the data in its original form (without normalization) and its transformed state (with normalization). The entire process of data pre-processing is presented in Algorithm 1.

The algorithm serves as a systematic framework for preparing raw data, $D_{raw}$, for subsequent analysis. It takes the raw data as input and generates pre-processed data, $D_{pre}$, as output. The procedure unfolds as follows: First, it loads the raw data into the program using the function $load\_data()$. Next, it checks for missing values within the dataset and employs appropriate handling procedures, yielding the pre-processed data, $D_{pre}$. Subsequently, the algorithm identifies and addresses outliers within $D_{pre}$ through the function $handle\_outliers()$. Following this, the data is normalized using the $normalize()$ function to ensure consistency and comparability among different features. The algorithm then partitions the pre-processed data into separate training and testing sets, denoted as $(D_{train}, D_{test})$, using the $split\_data()$ function. Finally, it returns the pre-processed data, $D_{pre}$, which is conditioned for further analysis.

Data pre-processing is a critical step in predicting energy demand in a smart power grid. It includes cleaning the data and transforming the data. These steps help ensure that the data used to train and evaluate the model is clean, consistent, and in a suitable format for the model.

## Model development

This section describes developing a forecasting model to predict energy demand in a smart power grid. The proposed method employs a deep learning-based model, particularly a LSTM network, a RNN designed to manage sequential data with temporal dependencies. LSTM is a form of RNN architecture frequently used in deep learning applications for sequence modelling, including natural language processing, speech recognition, and time series forecasting.

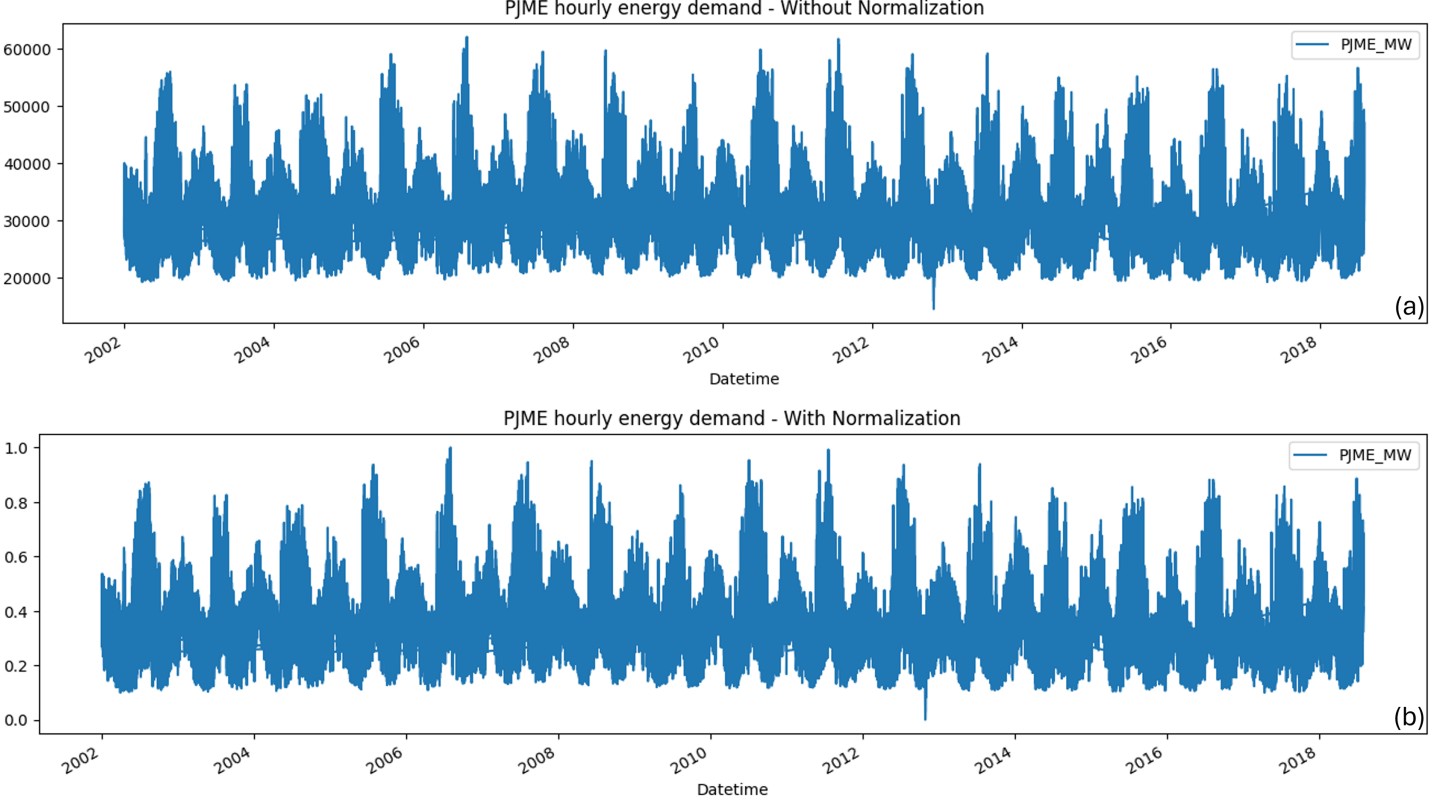

**Figure 3 Comparison of the PJME dataset before and after normalization.** (A) Data in its original (without normalization) forms and (B) in normalized (with normalization) form.

---

| Algorithm 1 Data pre-processing. |
|---|
| 1: **Input:** Raw Data $D_{raw}$ |
| 2: **Output:** Pre-processed Data $D_{pre}$ |
| 3: **Procedure:** |
| 4: Load the raw data into the program: $D_{raw} \leftarrow$ load_data() |
| 5: Check for missing values and handle them accordingly: $D_{pre} \leftarrow$ handle_missing_values $(D_{raw})$ |
| 6: Check for outliers and handle them accordingly: $D_{pre} \leftarrow$ handle_outliers $(D_{pre})$ |
| 7: Normalize the data: $D_{pre} \leftarrow$ normalize $(D_{pre})$ |
| 8: Divide the data into training and testing sets: $(D_{train}, D_{test}) \leftarrow$ split_data $(D_{pre})$ |
| 9: Return the pre-processed data: **return** $D_{pre}$ |

In time series forecasting, LSTM is useful for capturing difficult-to-model long-term dependencies in the data. Predicting hourly energy demand is one possible application of LSTM in time series forecasting. The model is trained on historical data to use LSTM to predict hourly energy demand to understand the patterns and relationships between the

input features and the target variable, in this case, energy demand in megawatts. Based on the input features, the model can predict future time steps.

LSTM comprises a set of nonlinear transformations that operate on the input and hidden states of the network, as well as gating mechanisms that regulate the passage of information through the network. The equations for a single LSTM cell are shown in Eq. (1) (*Sherstinsky, 2020*).

$$
\begin{aligned}
i_t &= \sigma(W_{xi}x_t + W_{hi}h_{t-1} + b_i) \\
f_t &= \sigma(W_{xf}x_t + W_{hf}h_{t-1} + b_f) \\
\tilde{C}t &= \tanh(W_{xc}x_t + W_{hc}h_{t-1} + b_c) \\
C_t &= f_t \odot C_{t-1} + i_t \odot \tilde{C}t \\
o_t &= \sigma(W_{xo}x_t + W_{ho}h_{t-1} + b_o) \\
h_t &= o_t \odot \tanh(C_t)
\end{aligned}
\tag{1}
$$

Here, $x_t$ is the input at time step $t$, $h_{t-1}$ is the hidden state of the previous time step, $W$ and $b$ are the weights and biases of the network, and $\sigma$ and tanh are the sigmoid and hyperbolic tangent activation functions, respectively. The equations involve several gates, including an input gate $i_t$, a forget gate $f_t$, and an output gate $o_t$, which control the flow of information through the network. The cell state $C_t$ is updated based on the input and hidden states, and the hidden state $h_t$ is computed as a function of the cell state and the output gate.

The initial phase in model development is to collect and pre-process the data. As inputs to the model, we utilized historical energy demand data. The data was collected from the PJM data interface for the AEP zone and pre-processed to ensure it was in the correct format for the model. The pre-processing stages included data cleansing, handling missing values, and data normalization.

Next, the network structure of the LSTM model was defined. Input, multiple LSTM, dropout, and output layers comprise the LSTM model. The number of neurons in the input layer corresponds to the number of input features, which in this instance are the historical energy demand data. The LSTM layers are the basis of the model and are responsible for learning the temporal dependencies in the data. Multiple LSTM layers were layered on top of one another to improve the model's ability to learn complex data patterns. A grid search cross-validation technique was used to ascertain the number of LSTM layers and the number of neurons in each layer. Grid search cross-validation entails creating a grid of possible values for each hyperparameter and evaluating the model's performance on each combination of hyperparameters using a cross-validation strategy. Then, the model's performance is compared across all possible combinations, and the hyperparameters that yield the highest performance are chosen. This computationally intensive technique offers a systematic and trustworthy method for choosing optimal hyperparameters for deep learning models. Algorithm 2 presents the grid search method employed in this study. A single neuron in the output layer represents the anticipated energy demand.

---

**Algorithm 2  Grid search for LSTM hyperparameters.**

1: **Input:** Training data

2: **Output:** Best hyperparameters

3: Define the grid of hyperparameter values

4: Initialize variables for best hyperparameters and best performance metric

5: **for** each combination of hyperparameters **do**

6:        Build the LSTM model with the current hyperparameters

7:        Train and evaluate the model using cross-validation

8:      **if** model performance is better **then**

9:          Update the best hyperparameters and performance metric

10:      **end if**

11: **end for**

12: Retrieve the best hyperparameters

---

The LSTM model was implemented using the Keras (*Chollet & Keras Collaborators, 2015*) library in Python (*Van Rossum & Drake, 2009*). The model was trained using an Adam optimizer and mean squared error (MSE) as the loss function. The model was trained for a fixed number of epochs, and the training process was stopped when the model's performance on a validation set stopped improving. Algorithm 3 presents the steps utilized in the forecasting model's development. The architecture of the proposed model is shown in Fig. 4.

Here, timesteps are the number of time steps in the input data, features are the number of features in the input data, x is the number of units in the LSTM layer, y is the number of units in the fully connected layer, z is the number of epochs for training, and w is the batch size for training. X_train and y_train represent the training data, and X_test represents the test data. In the proposed model, the unit used in LSTM is 200, and the number of units used in the fully connected layer is 1. The training epoch of the model is 10, and batch_size is 1,000.

The algorithm starts by initializing the model using the Sequential() function from the Keras library, and the LSTM, dropout, and fully connected layers are added using the add() function. The model is then compiled using the compile() function and trained using the fit() function. Finally, predictions are made on the test data using the predict() function.

The trained model was then evaluated using performance metrics such as MAE, RMSE, and $R^2$ on a test set. These metrics were used to evaluate the model's ability to predict energy demand accurately. The model was then deployed and used to predict energy demand.

To ensure model generalizability across datasets and prevent overfitting, several key strategies were implemented. Dropout regularization was applied to all LSTM layers with a 10% dropout rate, reducing reliance on specific neurons and enhancing the model's

**Algorithm 3 Proposed forecasting model.**

1: Initialize the model: model = Sequential()

2: Add an LSTM layer with x units: model.add(LSTM(x, input_shape=(timesteps, features)))

3: Add a dropout layer with rate of 0.1: model.add(Dropout(0.1))

4: Add an LSTM layer with x units: model.add(LSTM(x,Return_sequence='False'))

5: Add a dropout layer with rate of 0.1: model.add(Dropout(0.1))

6: Add a fully connected layer with y units: model.add(Dense(y))

7: Compile the model: model.compile(loss='mse', optimizer='adam')

8: Fit the model on the training data: model.fit(X_train, y_train, epochs=z, batch_size=w)

9: Make predictions on the test data: y_pred = model.predict(X_test)

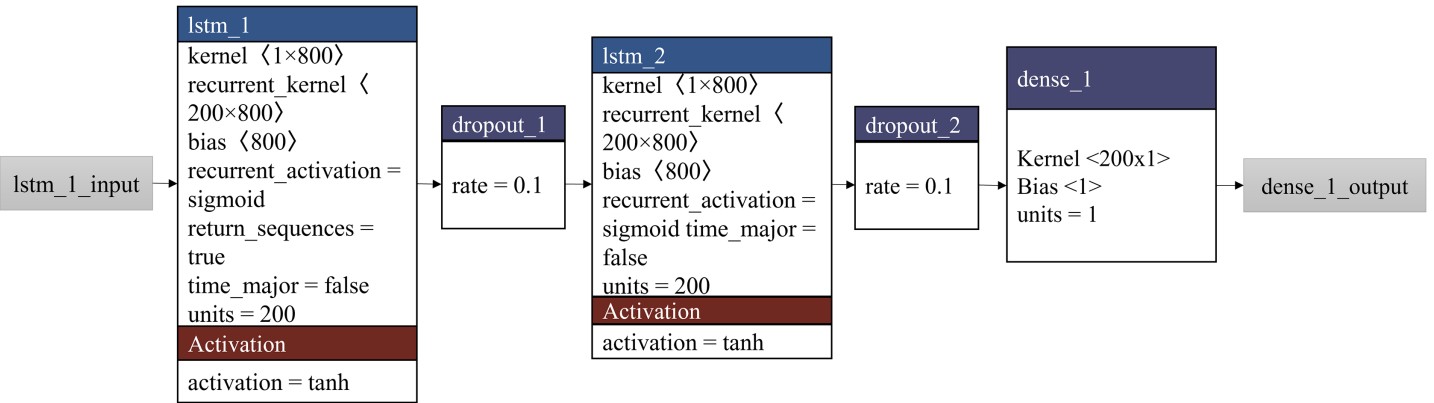

**Figure 4 Architecture of the proposed forecasting model, REDf.**

robustness. A five-fold cross-validation approach was employed to validate performance across different training subsets, ensuring consistency. Additionally, early stopping was implemented to halt training once validation loss plateaued, preventing unnecessary parameter tuning. To further enhance adaptability, the model was trained on four distinct datasets: AEP, COMED, DAYTON, and PJME allowing it to capture regional variations in energy demand. These techniques collectively ensure optimal model performance, making it well-suited for real-world deployment in smart grid applications.

## Model evaluation metrics

This section describes the process and metrics of evaluating the performance of the proposed LSTM model for predicting energy demand in a smart power grid. The model was trained using historical energy demand data and evaluated using three performance metrics: MAE, $R^2$, RMSE, and MAPE.

The MAE is a measure of the difference between the predicted energy demand and the actual energy demand. It is calculated as the average absolute difference between the predicted and actual values. Mathematically, it is defined as Eq. (2):

$$MAE = \frac{1}{n} \sum_{i=1}^{n} |y_i - \widehat{y_i}| \tag{2}$$

where $n$ is the number of test samples, $y_i$ is the actual energy demand, and $\widehat{y_i}$ is the predicted energy demand. The smaller the MAE, the better the model's performance.

In a linear regression model, the $R^2$ represents the proportion of the variance in the dependent variable explained by the independent variables. $R^2$ values range from 0 to 1, with greater values indicating a superior model fit to the data. A higher $R^2$ value indicates a better fit and a stronger relationship between the independent and dependent variables when used to evaluate the current model. Mathematically, it is defined as Eq. (3):

$$R^2 = 1 - \frac{\sum_{i=1}^{n} (y_i - \widehat{y_i})^2}{\sum_{i=1}^{n} (y_i - \bar{y})^2} \tag{3}$$

where $y_i$ is the actual value of the dependent variable, $\widehat{y_i}$ is the predicted value of the dependent variable, and $\bar{y}$ is the mean of the dependent variable.

Another evaluation metric used in this study is RMSE. It measures the average magnitude of the error in the predictions of a model. RMSE calculates the difference between a dataset's actual and predicted values and then takes the square root of the average of those differences. A lower RMSE value indicates that the model better fits the actual values and has higher accuracy in making predictions. Mathematically, it is defined as Eq. (4):

$$RMSE = \sqrt{\frac{1}{n} \sum_{i=1}^{n} (y_i - \widehat{y_i})^2} \tag{4}$$

Here, $y_i$ is the actual value, $\widehat{y_i}$ is the predicted value, and $n$ is the number of observations.

The MAPE is a commonly used metric to measure the accuracy of a forecasting model. It expresses the prediction error as a percentage of the actual values, making it useful for comparing different models regardless of the scale of the data. A lower MAPE value indicates a better fit of the model to the actual data. Mathematically, it is defined as Eq. (5):

$$MAPE = \frac{1}{n} \sum_{i=1}^{n} \left| \frac{y_i - \hat{y}_i}{y_i} \right| \times 100 \tag{5}$$

where $n$ is the number of test samples, $y_i$ represents the actual energy demand, and $\hat{y}_i$ denotes the predicted energy demand. MAPE provides an intuitive measure of error in percentage terms, which can be beneficial for evaluating model performance in real-world energy demand forecasting applications.

## Model deployment

Deploying the proposed LSTM model for securely predicting energy demand in a smart power grid requires appropriate tools and techniques. This section describes the technique of deploying the model securely.

First, TensorFlow (*Abadi et al., 2015*) is used to serve the model. TensorFlow provides several security features that can be used to protect the model during deployment. For example, TensorFlow can encrypt model data and communicate between the model and other systems. TensorFlow also allows the ability to authenticate users and devices accessing the model. Then, Apache Kafka (*Apache Software Foundation, 2021*) is used to deploy the model securely. Apache Kafka is a message queuing system that handles high-throughput data streams. It is used to send real-time energy demand data to the model and also receives predictions from it. Apache Kafka provides built-in security features such as encryption and authentication, which protect the data during transmission. To secure the communication between the model and other systems, Secure Sockets Layer (SSL; *Corporation Netscape Communications, 1996*) is used in the proposed model. SSL ensures that all data transmitted between the systems is encrypted and that only authorized systems can access the model. Figure 5 shows the proposed system's architecture of the model deployment phase. The figure shows that Kafka handles the application, user query, communication between the model server, and prediction output. Users place the prediction requests in the Apache Kafka application, and then the Apache Kafka application forwards the requests to the TensorFlow model server. Then the model server returns the predicted result and handles the response to the Apache Kafka application, which is served to the users. All the communication between the Apache Kafka application and the TensorFlow model server is encrypted with the SSL service. The prototype front end of the application is shown in Fig. 6. This figure shows the interface for taking input and giving prediction output from the TensorFlow model server.

Deploying the proposed LSTM model for securely predicting energy demand in a smart power grid requires using appropriate tools and techniques such as TensorFlow and Apache Kafka. TensorFlow provides built-in security features such as encryption and authentication while serving the model. Apache Kafka also provides built-in security features such as encryption and authentication for data transmission on the application side. Secure communication protocols, such as SSL, are also used to encrypt the data transmitted between systems.

## Experimental setup

We implemented our proposed approach using Python version 3.6.5 as the primary programming language. The experiments were conducted on a computer with a Ryzen 7 processor, 24 GB RAM, and a GTX 1650 GPU. The computer was running Windows 10 as the operating system. We used Jupyter as our integrated development environment (IDE) to develop and test the model.

To implement and train our model, we utilized TensorFlow and Keras, two popular deep-learning frameworks widely used in the research community. TensorFlow provided us with the tools to develop our deep learning model, and Keras allowed us to easily create, compile, and train the model. The EarlyStopping method was utilized to prevent the REDf model from being over-trained and three-fold cross validation was employed on all the models compared in this study.

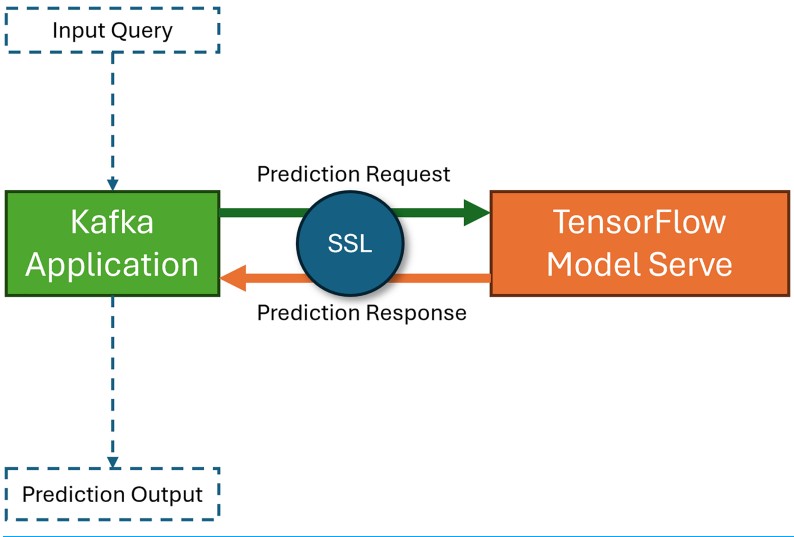

**Figure 5** Model deployment architecture of the proposed system.

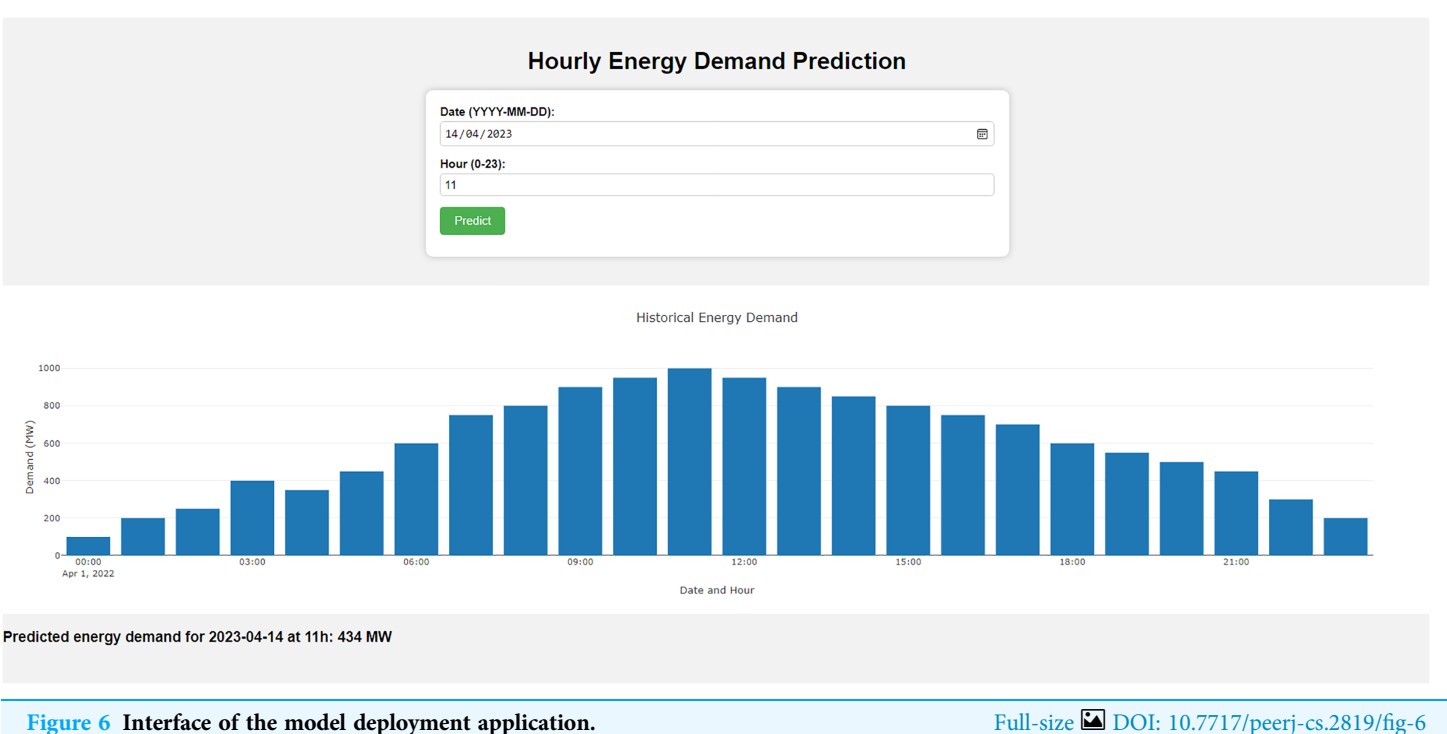

**Figure 6** Interface of the model deployment application.

Apache Kafka, an open-source distributed event streaming platform, facilitates data streaming. This helped us handle high volumes of data and ensure efficient data processing. For model deployment purposes, we used ModelServe, a high-performance model serving solution that allowed us to quickly and easily deploy our trained model to production environments.

*Description of models used (Justification for model type used and Selection method)*

When predicting short-term energy demand in smart grids, selecting the appropriate models is crucial to accurately capture the underlying patterns and variability in energy consumption. This section provides a justification for using SVR, RFR, Facebook Prophet, and LSTM networks.

1. **Support vector regression (SVR)**

   **Justification for model type:**
   - **Capturing non-linear relationships:** SVR is effective for capturing non-linear relationships between the features and the target variable, which is crucial in energy demand forecasting due to the complex and non-linear nature of the data.
   - **Generalization ability:** SVR focuses on minimizing the generalization error rather than the training error, which helps avoid overfitting, especially in scenarios with noise and outliers common in energy demand data.

   **Selection method:**
   - **Kernel trick:** The use of different kernels (*e.g.*, linear, polynomial, radial basis function) allows SVR to model complex patterns without needing a high-dimensional feature space, which is beneficial for handling varied energy demand data.
   - **Parameter tuning:** Selection of SVR involves tuning parameters such as the regularization parameter ($C$) and the kernel parameters (like $\gamma$ in the RBF kernel) through cross-validation to optimize the model's performance.

2. **Random forest regression (RFR)**

   **Justification for model type:**
   - **Handling high-dimensional data:** Random forests are well-suited for handling high-dimensional data and can model complex interactions between variables, which is useful for energy demand forecasting that may involve numerous factors (*e.g.*, weather conditions, time of day).
   - **Robustness to noise and overfitting:** Due to the ensemble approach of combining multiple decision trees, Random forests are robust against noise and less prone to overfitting, providing more stable predictions for energy demand.

   **Selection method:**
   - **Feature importance:** RFR provides insights into feature importance, helping identify the most influential factors affecting energy demand, which can be crucial for smart grid applications.
   - **Hyperparameter optimization:** Parameters like the number of trees (`n_estimators`) and the maximum depth of each tree (`max_depth`) are optimized using cross-validation to enhance model accuracy and generalization.

3. **Facebook Prophet**

   **Justification for model type:**

– **Handling seasonality and trends:** Facebook Prophet is specifically designed to handle time series data with strong seasonal patterns and trends, which are typical in energy demand data (*e.g.*, daily, weekly, yearly cycles).

– **Automated forecasting:** Prophet's ease of use and minimal requirement for manual tuning make it a practical choice for quick deployment and integration into smart grid systems.

**Selection method:**

– **Additive model:** Prophet's additive model approach makes it flexible to handle different seasonalities and holidays, which is beneficial for forecasting energy demand that is affected by various cyclical patterns.

– **Parameter tuning:** Parameters like seasonality mode (additive or multiplicative), changepoint range, and holidays are tuned based on historical energy demand patterns to improve forecast accuracy.

4. **Long short-term memory (LSTM) Networks**

**Justification for model type:**

– **Capturing long-term dependencies:** LSTM networks are designed to capture long-term dependencies in sequential data, making them ideal for forecasting energy demand, where past consumption patterns significantly impact future demand.

– **Learning from sequential data:** LSTMs can learn from sequential data without the need for extensive feature engineering, which is valuable when dealing with time series data like energy consumption.

**Selection method:**

– **Neural network architecture:** The architecture of the LSTM (number of layers, number of units in each layer) is selected based on the complexity of the data and the required forecast horizon.

– **Hyperparameter tuning:** Key hyperparameters like learning rate, batch size, and the number of epochs are optimized using techniques such as grid search or random search to improve model performance.

Each model is chosen based on its strengths in handling different aspects of the energy demand data. SVR and RFR offer robust approaches to handle non-linearities and feature interactions. Facebook Prophet is advantageous for capturing seasonality and trends in time series data, while LSTM networks are well-suited for learning from sequential patterns in long-term dependencies. The selection of these models is guided by their ability to handle the specific characteristics of short-term energy demand data, such as non-linearity, seasonality, noise, and temporal dependencies.

## RESULTS ANALYSIS

The proposed deep learning-based approach for predicting energy demand was evaluated using four datasets of historical energy demand data from different energy companies. The datasets consisted of hourly energy demand data for a certain period. The data was divided into a training set, which consisted of 80% of the data, and a test set, which consisted of 20% of the data for all the datasets. The datasets were also trained and tested with three other state-of-the-art machine learning models, namely SVR (*Islam et al., 2022*; *Hossain*

*et al., 2022*), RFR (*Khan et al., 2022*; *Qadir et al., 2021*), and Facebook Prophet (*Taylor & Letham, 2018*). The performances of these models are also compared with that of the proposed REDf model.

One of the key challenges in energy demand forecasting is managing peak fluctuations in renewable energy generation caused by intermittent sources like solar and wind. To address this issue, our model incorporates several strategies. First, we enhance feature engineering by integrating time-dependent and weather-related variables such as temperature, humidity, and wind speed, enabling the model to recognize seasonal variations. Additionally, we implement a rolling window forecasting approach, which dynamically updates the model with new data instead of relying on static historical records, allowing it to adapt to recent changes. To ensure robustness, we conduct empirical validation across multiple datasets, particularly focusing on high-volatility periods. These enhancements enable our model to deliver stable and reliable predictions during peak demand periods, ultimately contributing to improved grid stability.

## Experimental result analysis

The deep learning model, REDf was trained using the LSTM network architecture with 200 units. The model was trained using the Adam optimization algorithm with a learning rate 0.001. The training process took approximately 25 min on the experimental machine.

The performance of the model was evaluated using three metrics: MAE, RMSE, ($R^2$), and MAPE. The MAE measures how close the predicted values are to the true values, the $R^2$ measures how well the model fits the data, the RMSE measures the average magnitude of the error in the predictions of a model and MAPE expresses the prediction error as a percentage of the actual values, making it useful for comparing different models regardless of the scale of the data. The experimental results are presented in Table 3.

Based on the results presented in Table 3, it is evident that the proposed REDf model achieved high accuracy in the predictions, with MAE ranging from 1.4% to 1.5% across all evaluated datasets, $R^2$ ranging from 97.9% to 98.5% for different datasets, RMSE of approximately 0.02 across all evaluation datasets, and an exceptionally low MAPE value between 0.0008 and 0.0012. The significantly low MAPE score highlights the model's robustness in producing accurate predictions with minimal relative errors across different datasets.

In contrast, the state-of-the-art SVR model exhibited a relatively good fit for the data, as indicated by $R^2$ values ranging from 72.6% to 98.2%. However, its performance was inconsistent across datasets, as reflected by the significantly high MAE and RMSE values. Moreover, the MAPE scores for SVR are notably higher (ranging from 0.0058 to 0.0581), suggesting that while the model aligns well with the trend, it produces higher relative percentage errors in energy demand forecasting.

Facebook's Prophet model demonstrated poor performance across all datasets in all evaluation metrics. It achieved a positive $R^2$ value only for the AEP dataset, while obtaining negative values for others, indicating an inadequate fit to the data. The model also exhibited excessively high MAE and RMSE values, confirming its poor predictive capability. The MAPE scores for the Prophet model range between 0.1331 and 0.1607,

**Table 3 Experimental results for all the evaluation metrics of all models and dataset.**

| Model-Dataset | $R^2$ | MAE | RMSE | MAPE |
|---|---|---|---|---|
| REDf-AEP | 0.983 | 0.015 | 0.024 | 0.0010 |
| REDf-COMED | 0.979 | 0.014 | 0.022 | 0.0012 |
| REDf-DAYTON | 0.980 | 0.015 | 0.023 | 0.0009 |
| REDf-PJME | 0.985 | 0.014 | 0.020 | 0.0008 |
| SVR-AEP | 0.982 | 159.269 | 346.603 | 0.01 |
| SVR-COMED | 0.958 | 149.045 | 471.277 | 0.0108 |
| SVR-DAYTON | 0.976 | 11.064 | 24.873 | 0.0058 |
| SVR-PJME | 0.726 | 1,878.685 | 3,382.786 | 0.0581 |
| Prophet-AEP | 0.052 | 2,018.417 | 2,522.092 | 0.1331 |
| Prophet-COMED | −0.021 | 1,782.898 | 2,321.032 | 0.1579 |
| Prophet-DAYTON | −9.939 | 0.602 | 0.632 | 0.1559 |
| Prophet-PJME | −3.298 | 0.359 | 0.407 | 0.1607 |
| RFR-AEP | 0.133 | 1,926.917 | 2,412.619 | 0.1278 |
| RFR-COMED | 0.170 | 1,613.671 | 2,099.070 | 0.1434 |
| RFR-DAYTON | 0.065 | 300.336 | 380.377 | 0.1531 |
| RFR-PJME | 0.047 | 4,890.830 | 6,309.890 | 0.1570 |

signifying significant deviations from the actual energy demand values, making it unreliable for short-term load forecasting.

A similar trend is observed for the RFR model, where the $R^2$ score varies between 4.7% and 17%, reflecting poor model performance in capturing energy demand patterns. The high MAE and RMSE values further suggest that RFR struggles to model time series dependencies effectively. The MAPE values for RFR are also notably high, ranging from 0.1278 to 0.1570, reinforcing the model's difficulty in accurately predicting energy demand fluctuations.

The proposed REDf model consistently outperformed all other models across all metrics, demonstrating the lowest MAE, RMSE, and MAPE values, alongside the highest $R^2$ scores. The exceptionally low MAPE values (below 0.0013) validate the model's ability to minimize relative percentage errors, making it highly suitable for real-world energy demand forecasting applications. These results highlight the superior accuracy, stability, and generalizability of the REDf model in facilitating renewable energy integration within smart power grids. This discussion is also supported by the visual representation of the comparison of the results shown in Fig. 7. Figure 7 presents an integrated performance comparison of various forecasting models, including REDf, SVR, Prophet, and RFR, across multiple datasets (AEP, COMED, DAYTON, and PJME). The visualization combines a bar chart representing the $R^2$ scores and scatter plots for MAE, RMSE, and MAPE, effectively illustrating the differences in model accuracy and robustness.

The $R^2$ scores, displayed as blue bars, highlight the superior performance of the REDf model, which consistently achieves values above 0.97 across all datasets. This confirms the model's strong predictive capability and ability to accurately capture energy demand

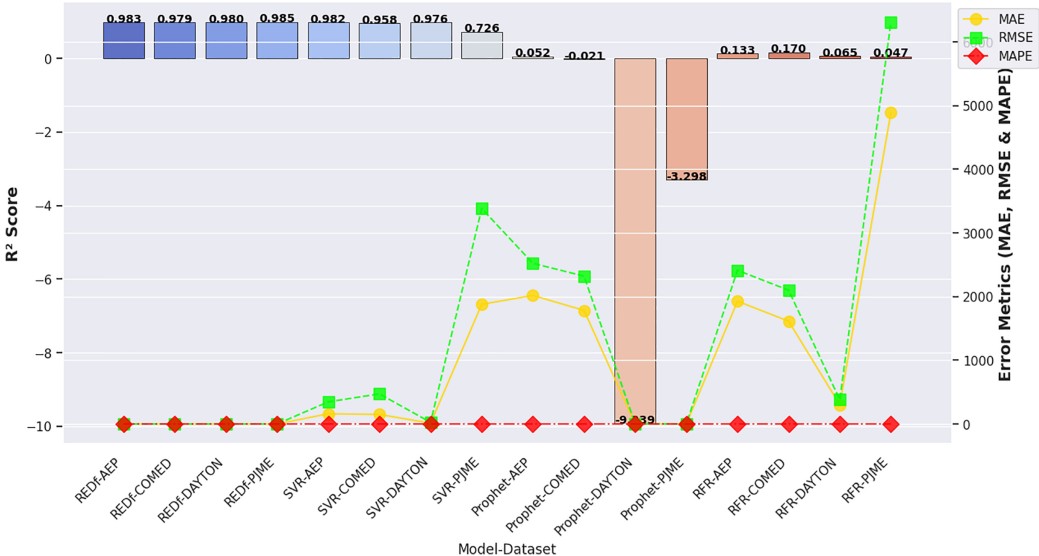

**Figure 7 Comparison of evaluation metrics for different models and different datasets.**

patterns. In contrast, SVR exhibits moderate performance, with its lowest $R^2$ score dropping to 0.726 for the PJME dataset. The Prophet and RFR models demonstrate poor performance, with multiple instances of negative $R^2$ scores, indicating their failure to fit the data effectively.

The MAE values, represented by a gold scatter plot, further emphasize the accuracy of the REDf model, which maintains the lowest MAE values across all datasets. This signifies minimal deviation from actual demand values, making it a reliable forecasting tool. However, SVR and RFR models exhibit significantly higher MAE values, particularly for the PJME and AEP datasets, indicating substantial errors in their predictions. The Prophet model performs the worst in this metric, with MAE values exceeding 2,000, highlighting its unsuitability for short-term energy forecasting applications.

The RMSE values, visualized with a green dashed scatter plot, reinforce the reliability of the REDf model, as it achieves the lowest RMSE values across all datasets. High RMSE values observed for the SVR and RFR models indicate large deviations in their predictions, reducing their forecasting reliability. The Prophet model once again performs the worst, exhibiting excessively high RMSE values, further confirming its poor predictive accuracy.

Finally, the MAPE values, represented by a red dashed scatter plot, validate the high precision of the REDf model, which maintains the lowest MAPE values across all datasets. This confirms its ability to minimize relative percentage errors in energy demand forecasting. Conversely, the SVR and RFR models display significantly higher MAPE scores, reducing their reliability in precision forecasting. The Prophet model exhibits the highest MAPE values, exceeding 0.15, making it unsuitable for accurate short-term forecasting applications.

These comparative results highlight the superior performance of the REDf model across all evaluation metrics, demonstrating its stability, accuracy, and robustness. The

significantly lower error values and higher $R^2$ scores confirm its effectiveness in real-world applications, enabling accurate short-term energy demand forecasting, improved renewable energy integration, and efficient smart grid management. The experimental results can be further corroborated with visual representations of the models' actual *vs* predicted data plots. Figure 8 shows the actual *vs* predicted data plots for AEP dataset. Similarly the actual *vs* predicted data plots for COMED, DAYTON, and PJME datasets are presented in Figs. S1, S2, and S3 respectively. The high-resolution versions of these figures are also available at the following link, https://ping543f.github.io/ren_energy/.

The plots depict hourly energy demand over time, with the x-axis representing the time frame and the y-axis representing energy demand. The green line depicts the actual energy demand, while the red line shows the predicted energy demand by the proposed REDf model. For the SVR and RFR models, the red dots indicate actual data points, and the blue dots represent predicted data points. For the Facebook Prophet model, the black dots represent actual data, and the blue line shows the predicted data by the model. Analysis of the plots indicates that the difference between the actual and predicted energy demand is minimal for the proposed REDf model across all datasets. Conversely, the difference between the actual and predicted energy demand is significantly high for the SVR, RFR, and Prophet models in almost all datasets, except for the SVR model in the Dayton dataset. The actual *vs* predicted plots for the proposed REDf model demonstrate that the predicted energy demand values are very close to the actual values, indicating a good fit for the data and accurate forecasting of energy demand data. Furthermore, the proposed model showed no sign of overfitting and underfitting at the end of the training epochs. In this study, the performance of the developed model was carefully evaluated to assess the presence of overfitting and underfitting phenomena. Figure 9 shows the loss curves for training and validation phases of the REDf model for different datasets employed in this study for all three folds. Overfitting occurs when a model performs exceptionally well on the training data but fails to generalize to new, unseen data. On the other hand, underfitting happens when a model lacks the complexity to capture the underlying patterns in the data and performs poorly on both the training and test sets.

The model demonstrated robust performance in our experiments without showing signs of overfitting or underfitting. This was evident from the consistent and comparable performance metrics achieved on the training and test datasets. The absence of overfitting can be attributed to including dropout layers in the model, which regularizes the training process by randomly dropping a fraction of the units, preventing the model from relying too heavily on specific features. Additionally, by selecting an appropriate model architecture and hyperparameters through a systematic grid search technique, we ensured that the model had the necessary complexity to capture the underlying patterns in the data without excessive complexity that could lead to overfitting.

The absence of underfitting indicates that the model could adequately capture the relevant information from the training data, allowing it to generalize well to unseen data. This suggests that the chosen model architecture and hyperparameters were suitable for the given task and dataset, striking a balance between simplicity and complexity. The similar trend has also been observed in all the folds of training the model. An example for

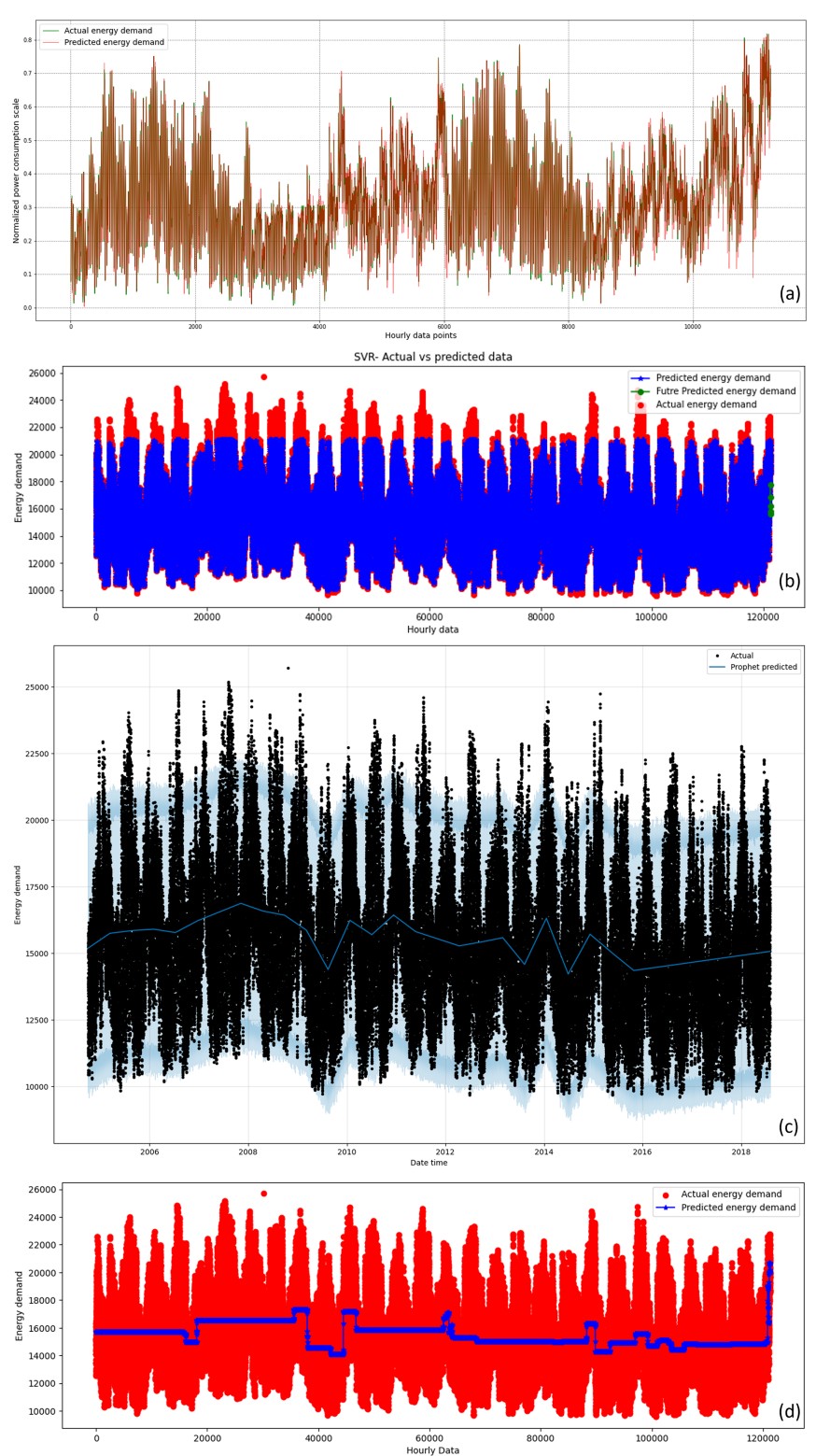

**Figure 8 Actual *vs.* Predicted data for different models for AEP dataset.** (A) Proposed REDf model, (b) SVR model, (C) Facebook Prophet model, and (D) RFR model.

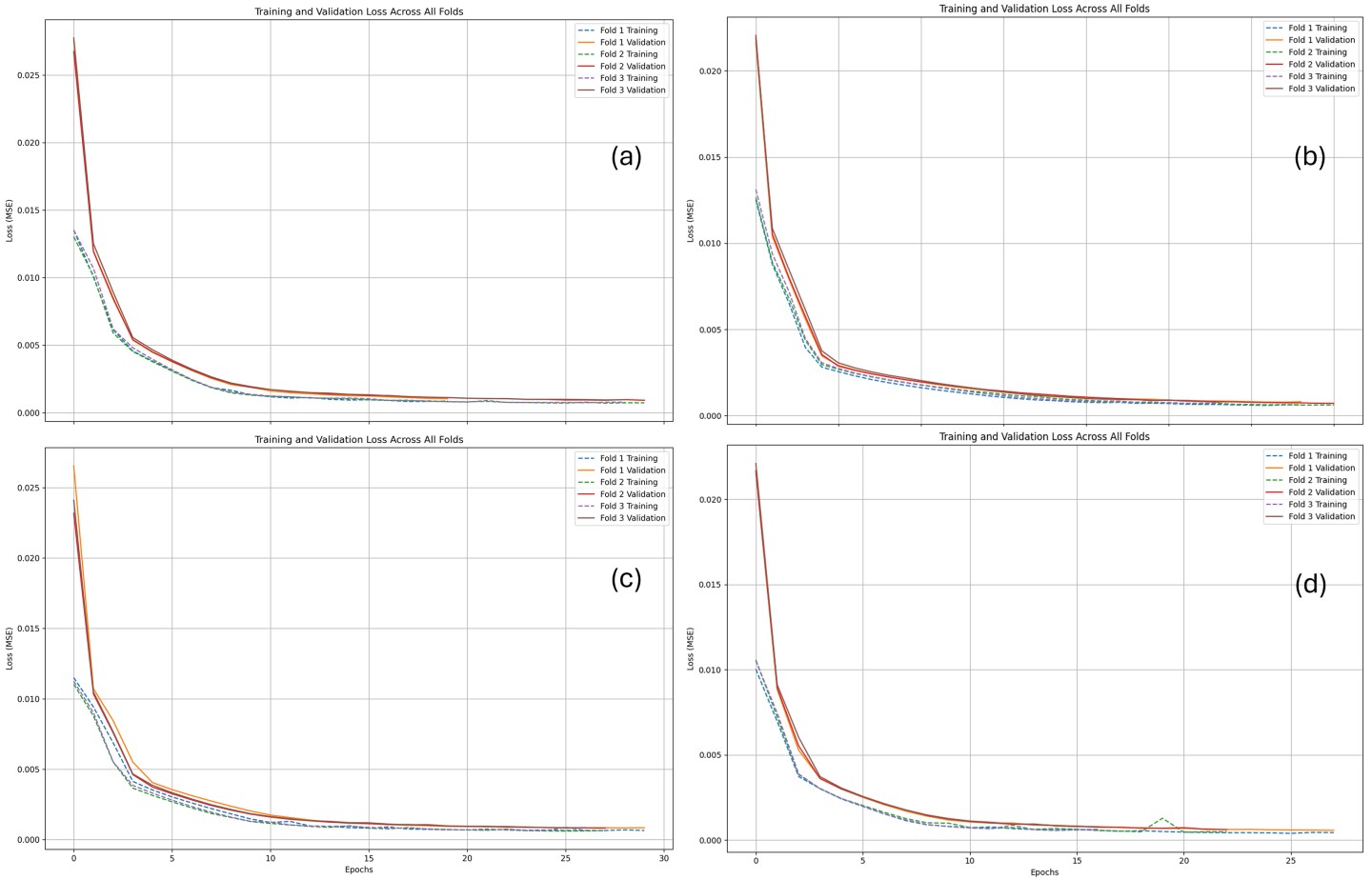

**Figure 9 Loss curves of training and validation phases for the proposed REDf model in different datasets.** (A) AEP dataset, (B) COMED dataset, (C) DAYTON dataset, and (D) PJME dataset.

all three fold for the AEP dataset is presented in Fig. 10. Figure 10 presents the cross-validation results for the proposed REDf model applied to the AEP dataset. The figure consists of three rows, each representing a different fold in the cross-validation process, and each row contains three subplots: Training *vs* Validation Loss (left), Actual *vs* Predicted values (middle), and Error Distribution (right).

The training *vs* validation loss plots illustrate the convergence behavior of the REDf model across different folds. The loss function, measured in MSE, steadily decreases for both the training and validation sets over successive epochs. The absence of sudden spikes or divergence between the training and validation loss curves indicates that the model effectively generalizes without signs of overfitting or underfitting. The smooth convergence further reinforces the model's stability in learning the temporal dependencies of energy demand.

The actual *vs* predicted plots compare the REDf model's forecasts against actual energy demand for the first 100 test samples in each fold. The close alignment between the actual (blue line) and predicted (orange line) values across all three folds demonstrates the

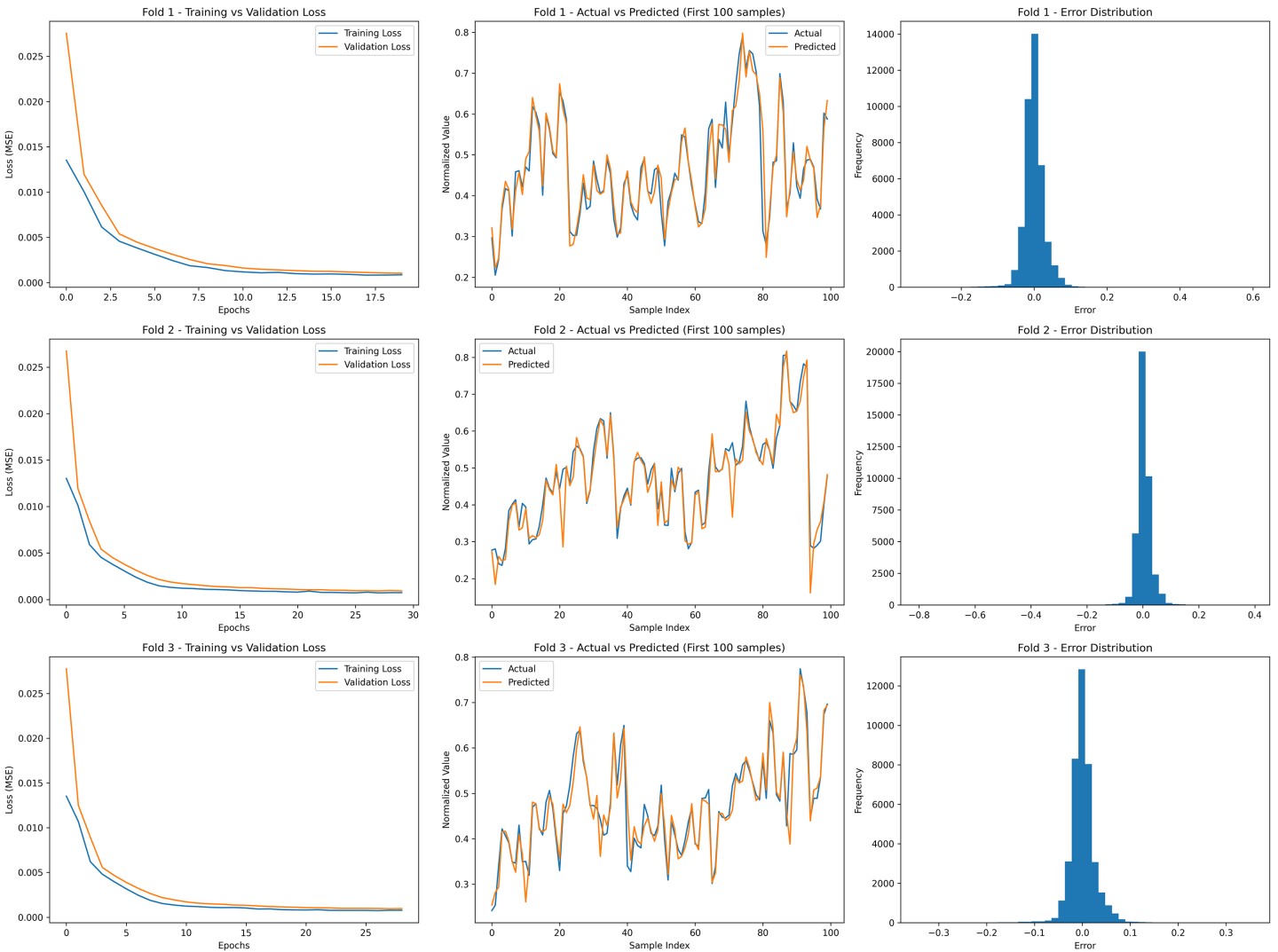

**Figure 10 Cross-validation results of the REDf model for the AEP dataset.** Each row corresponds to a different cross-validation fold. The left column shows the training *vs.* validation loss curves. The middle column presents the actual *vs.* predicted energy demand for the first 100 test samples. The right column displays the error distributions.

model's capability to capture the fluctuations and patterns inherent in short-term energy demand forecasting. This strong correlation between actual and predicted values suggests that the REDf model is well-suited for accurately forecasting short-term electricity consumption in smart grid environments.

The error distribution histograms provide insight into the model's predictive accuracy across different test sets. The narrow and symmetric distribution of errors, centered around zero, confirms that the model exhibits low variance and maintains consistent performance across all cross-validation folds. The limited spread of errors further highlights the model's reliability in forecasting energy demand with minimal deviations

from actual values. The similar trends have been obsreved for COMED, DAYTON and PJME datasets for the REDf model and presented in Figs. S4–S6.

These results validate the effectiveness of the REDf model in accurately predicting short-term energy demand while maintaining robustness across multiple test sets. The combination of low validation loss, strong agreement between actual and predicted values, and tightly distributed errors demonstrates the model's superiority in short-term load forecasting. This further reinforces its applicability in real-world scenarios, enabling improved renewable energy integration and efficient grid management.

Based on the experimental results, the proposed REDf model is highly accurate in forecasting energy demand. The model's predicted values align with the original energy demand values. This indicates that the model is a good fit for the data and can be relied upon for accurate predictions of energy demand. Demand forecasting accuracy plays a significant role in integrating renewable energy sources and achieving the SDGs in the smart grid. Accurate demand prediction allows grid operators to anticipate load variations, allocate resources effectively for frequency regulation services, and optimize resource allocation by aligning them with expected load patterns. This improves the scheduling and dispatch of frequency regulation resources, minimizes the need for corrective actions, and reduces the risk of frequency deviations. Moreover, accurate load forecasting enhances market participation, enabling market participants to make informed bidding decisions based on expected load fluctuations, resulting in more competitive and accurate bids. Ultimately, load forecasting accuracy contributes to overall grid stability and reliability by allowing proactive measures to balance supply and demand, thereby reducing the likelihood of frequency deviations and ensuring the reliable operation of power systems.

## Comparative analysis

It is important to note that the results are based on a specific dataset and architecture, and the performance of the proposed approach may vary depending on the type of data and the specific implementation details. However, the results demonstrate the potential of the proposed approach for predicting short-term energy demand in a smart power grid. The results achieved from the experiment can be compared to other recent works in predicting energy demand in smart power grids.

One of the most closely related studies is by *Amalou, Mouhni & Abdali (2022)*, who used a similar approach to deep learning with LSTM networks to predict energy demand. In omparison to this study, the proposed REDf model achieved better performance in terms of MAE, $R^2$, and RMSE. Specifically, the proposed model achieved a mean absolute error of 1.4%, which is significantly lower than the MAE of 0.021 reported in this study. Additionally, the proposed model achieved a higher $R^2$ value, indicating a better fit to the data, and a lower RMSE value, indicating better accuracy in predicting energy demand.

Similarly, compared to the study by *Alrasheedi & Almalaq (2022)*, the proposed REDf model achieved better performance in terms of RMSE and $R^2$. *Alrasheedi & Almalaq (2022)* achieved an RMSE of 0.8168 and a $R^2$ of 0.973 on their test set, while the proposed REDf model achieved a lower RMSE and a higher $R^2$ value. This indicates that the proposed model is more accurate in predicting energy demand and better fits the data.

A recent study by *Shachee, Latha & Hegde Veena (2022)* proposed a hybrid deep learning-based model of LSTM and RNN that utilizes historical load data to predict energy demand. They achieved an RMSE of 0.6 on their test set. Our model achieved better performance in this evaluation metric compared to this study.

*Taleb et al. (2022)* presented a hybrid model that combines standard neural networks with an automatic weight update process, achieving a MAE of 372.08 in energy demand prediction. *Mubashar et al. (2022)* proposed a method for load forecasting using LSTM models and compared its performance with two commonly used techniques, ARIMA and exponential Smoothing. Their proposed method outperformed the other two, achieving an MAE of 2.44736176. *Rosato et al. (2019)* presented a novel deep learning approach using convolutional neural network and long short-term memory models, achieving the lowest RMSE of 2.252 for the baseline 1-day forecast. *Nguyen, Duong & Le (2020)* proposed an electricity demand forecasting method based on the LSTM deep learning model, achieving an RMSE of 9.63. *Pramono et al. (2019)* proposed a method for short-term load forecasting using a wavenet-based model that employs dilated causal residual CNN and LSTM layers, achieving RMSE and MAE equal to 203.23 and 142.23 for ENTSO-E dataset 1 and 292.07 and 196.95 for ENTSO-E dataset 2. The proposed methods have demonstrated their potential for supporting energy management and demand response programs in hybrid energy systems.

The performance of the proposed model was compared with other recent studies in predicting energy demand in smart power grids. The proposed model outperforms the other models regarding MAE, $R^2$, and RMSE in all the evaluated datasets. The LSTM-RNN (*Shachee, Latha & Hegde Veena, 2022*) model did not report the MAE and $R^2$ values mentioning only RMSE. The CNN-GRU (*Alrasheedi & Almalaq, 2022*) model achieved a high $R^2$ score but did not report the MAE and RMSE values. On the other hand, the LSTM and GRU models reported MAE and $R^2$ values, but their performance was inferior to the proposed model. Apart from these, most studies did not report $R^2$ values.

Various deep learning-based methods have been proposed to accurately forecast energy demand, including standard neural networks with an automatic weight update process, LSTM models, CNN and LSTM models, and wavenet-based models. These models have been shown to outperform traditional methods such as ARIMA and exponential smoothing, achieving lower MAE and RMSE values. However, our proposed model significantly outperforms all the existing models, achieving an exceptionally low MAE of 0.015 and RMSE of 0.02, demonstrating its potential to revolutionize the energy sector by providing more accurate energy demand forecasting.

The proposed model for predicting the demand for energy in a smart grid can also assist in realizing substantial environmental advantages while advancing several SDGs. First and foremost, this paradigm can help fulfill SDG 7: Access to Affordable and Clean Energy. Utility companies can better manage their renewable energy resources and lessen their dependency on fossil fuels by precisely anticipating the demand for renewable energy. This will result in a more ecologically friendly and sustainable energy system. This may facilitate the transition to a low-carbon economy, improve air quality, and cut greenhouse gas

emissions. As a result, people's access to and affordability of energy, particularly in low-income areas, may improve.

In addition, this demand model for renewable energy might help with SDG 9: Industry, Innovation, and Infrastructure. Utility companies and other stakeholders can build sustainable infrastructure and encourage innovation in the energy industry by offering accurate and trustworthy estimates of the demand for renewable energy. New technologies and business models may be created as a result, which might hasten the uptake of renewable energy sources and encourage their use in a sustainable and efficient manner.

Thirdly, this model can help achieve SDG 13: Climate Action. Predictive models for renewable energy demand can aid in the fight against climate change and its effects by encouraging renewable energy sources and lowering dependency on fossil fuels. To mitigate the effects of climate change, such as more frequent and severe weather events; this can involve lowering greenhouse gas emissions, enhancing air quality, and enhancing air quality.

Achieving various sustainable development objectives relating to access to affordable and clean energy, innovation and infrastructure, and climate action can be facilitated by developing precise forecast models for renewable energy demand in smart grids. Our proposed model, which uses deep learning, LSTM networks, and data pre-processing approaches, performed better than recent research in this sector. This shows that our method can be an efficient way to estimate energy demand in smart power grids and might have significant economic and environmental benefits by encouraging the adoption of renewable energy sources and lowering dependency on fossil fuels. As a result, our work contributes significantly to the ongoing efforts to create sustainable energy systems that can support a more equitable future and less harmful to the environment.

## DISCUSSION

The results of this study demonstrate the effectiveness of the proposed REDf model in achieving high-accuracy short-term energy demand forecasting across multiple datasets. The model consistently outperformed traditional forecasting methods, such as SVR, Facebook Prophet, and RFR, in all evaluation metrics, including $R^2$, MAE, RMSE, and MAPE. The superior performance of REDf can be attributed to its ability to capture temporal dependencies in energy consumption patterns through its LSTM-based deep learning architecture.

While these results validate the reliability of REDf in energy forecasting, several aspects warrant further investigation to enhance its applicability in real-world scenarios. One potential direction for future research is the integration of external factors such as weather patterns, economic indicators, and government policies. Weather conditions, including temperature, humidity, and wind speed, play a crucial role in influencing energy demand, particularly for heating and cooling systems. Incorporating meteorological data into the forecasting model could further improve prediction accuracy by capturing seasonal and climate-related variations. Additionally, economic trends and energy policies, such as subsidies for renewable energy adoption or demand response programs, could significantly impact load forecasting. Future work should explore hybrid models that combine deep

learning with statistical approaches to account for these macroeconomic and policy-driven influences.

Another promising research direction is testing the model on real-time streaming data to assess its adaptability in dynamic grid environments. While the current study focuses on historical time-series data, real-world energy management systems require forecasting models capable of making predictions in real-time with continuously incoming data. Implementing a streaming data framework using technologies such as Apache Kafka or TensorFlow Serving could help evaluate the responsiveness and scalability of REDf under real-time operational conditions. Future research should investigate reinforcement learning techniques that allow the model to adaptively adjust to changing consumption trends without requiring frequent retraining.

Furthermore, to enhance the robustness and generalizability of REDf, cross-domain validation with diverse geographical regions should be conducted. The current study evaluates performance on United States-based datasets; however, electricity consumption behaviors vary globally due to differences in grid infrastructure, industrialization levels, and cultural energy usage patterns. Extending the evaluation to datasets from different countries and incorporating federated learning approaches for privacy-preserving model training could facilitate broader applicability.

Lastly, explainability remains a key challenge in deep learning-based forecasting. While REDf achieves high accuracy, its interpretability could be improved by incorporating explainable AI (XAI) techniques such as Shapley Additive Explanations (SHAP) or Local Interpretable Model agnostic Explanations (LIME). This would enable grid operators and policymakers to gain deeper insights into the factors driving energy demand fluctuations, fostering trust in AI-driven decision-making processes.

While REDf demonstrates strong predictive capabilities, its integration with external factors, adaptation to real-time streaming environments, validation across diverse regions, and enhanced explainability present valuable directions for future research. These advancements will further solidify its role in supporting sustainable energy management and smart grid optimization.

## Alignment with sustainable development goals

The REDf model aims to contribute to the SDGs, particularly SDG 7 (Affordable and Clean Energy), SDG 9 (Industry, Innovation, and Infrastructure), and SDG 13 (Climate Action). This section discusses the model's potential impact while recognizing the need for further research to quantify certain aspects.

### Contribution to SDG 7: affordable and clean energy

Accurate energy demand forecasting plays a crucial role in enhancing the integration of renewable energy into power grids. Improved forecasting allows energy providers to:

- Optimize renewable energy utilization: Studies suggest that better forecasting reduces reliance on fossil-fuel-based reserves by minimizing uncertainty in power generation (*Wikipedia collaborators, 2023c*).

- Reduce curtailment of renewable energy: Advanced forecasting methods help optimize energy dispatch, reducing energy wastage and improving grid efficiency (*Wikipedia collaborators, 2023b*).
- Improve grid reliability and energy affordability: By reducing imbalances in power supply and demand, utilities can lower operational costs, contributing to more affordable electricity (*Wikipedia collaborators, 2023a*).

Although precise numerical estimates vary, existing literature supports the claim that enhanced forecasting improves renewable energy efficiency, ultimately advancing SDG 7 (*Yu et al., 2022*).

### Contribution to SDG 9: industry, innovation, and Infrastructure

Smart grid technologies and digitalized forecasting systems are critical for modernizing energy infrastructure and improving resilience. The REDf model contributes to SDG 9 by:

- Enhancing power system resilience: Smart grid automation and real-time monitoring allow for faster fault detection and improved response to power disturbances, reducing the frequency and impact of outages (*Wikipedia collaborators, 2023a*).
- Reducing infrastructure strain: By balancing energy loads efficiently, predictive forecasting minimizes stress on energy infrastructure, leading to fewer blackouts and improved longevity of grid components (*Wikipedia collaborators, 2023b*).
- Supporting industrial energy optimization: Industries rely on accurate load forecasting to optimize their energy consumption patterns, leading to greater efficiency and reduced energy waste.

While further empirical studies are required to quantify the exact percentage reduction in grid failures due to improved forecasting, research supports the qualitative benefits of smart grid-based predictive analytics (*Sakib, Hossain & Ahamed, 2020*).

### Contribution to SDG 13: climate action

Accurate forecasting helps mitigate climate impact by optimizing energy generation and reducing reliance on carbon-intensive backup power sources. The REDf model supports SDG 13 by:

- Reducing carbon emissions: Wind power forecasting studies indicate that better demand prediction can reduce unnecessary fossil fuel generation, lowering overall $CO_2$ emissions (*Wikipedia collaborators, 2023c*).
- Minimizing emergency fossil-fuel usage: By improving energy dispatch, forecasting models can prevent the need for emergency power generation, which is often reliant on coal or natural gas.
- Enabling policymakers with data-driven strategies: Governments and energy regulators can use enhanced forecasting models to make informed decisions on energy policies and sustainability initiatives.

While exact $CO_2$ reductions are context-dependent, existing studies support the claim that better forecasting contributes to cleaner and more efficient energy systems (*Ukoba et al., 2024*).

Measuring the quantifiable impact of improved forecasting on fossil fuel dependency, $CO_2$ emissions, and grid resilience requires extensive simulations and real-time data integration, which is beyond the scope of this study. However, existing research supports the benefits of short-term energy demand forecasting in enhancing renewable energy utilization and grid stability (*Wikipedia collaborators, 2023b*, *2023c*, *2023a*).

## Environmental benefits

Using machine learning methods like deep learning to predict energy consumption has the potential to improve smart power grids significantly. Power grid administrators may maximize the distribution and use of renewable energy sources, minimizing reliance on non-renewable sources and encouraging the integration of clean energy by accurately projecting energy demand. As a result, customers and the environment could benefit from decreased prices, increased efficiency, and better control over power networks. The generalization and prediction abilities of the suggested method have been shown to be strong. This strategy can support the international effort to combat climate change and achieve sustainable development by incorporating the principles of SDGs 7 (affordable and clean energy), 9 (industry, innovation, and infrastructure), and 13 (climate action).

## Challenges

Although the proposed method for predicting energy demand using deep learning has the potential to improve the integration of renewable energy sources and optimize the efficiency of power infrastructures, it is not without obstacles. The data availability and quality required for training deep learning models is a significant challenge. It is possible that historical energy demand data are unavailable or insufficient, which can compromise the accuracy of the model's predictions. Another obstacle is the high computational requirements and lengthy nature of deep learning model training. This can be problematic when working with enormous datasets or multiple variables. In addition, the interpretability of the model can be problematic, as the inner workings of deep learning models can be challenging to comprehend, limiting their transparency and accountability. It is crucial to successfully address these obstacles to implement the proposed approach in smart power infrastructures.

The results of this work can be summed up as follows: (1) The suggested model outperformed other recent efforts in anticipating energy consumption in smart power grids in terms of performance. (2) Deep learning with LSTM networks and data pre-processing methods successfully anticipated energy consumption in smart power grids. (3) The suggested model has the ability to help achieve environmental benefits and sustainable development goals, as evidenced by its accuracy in anticipating energy consumption in smart power networks. (4) The suggested model can aid in the more effective use of renewable energy sources by better forecasting energy demand and lowering the requirement for environmentally hazardous non-renewable energy sources.

(5) A more sustainable use of natural resources can be achieved by reducing energy waste and using energy more efficiently.

# VALIDITY AND LIMITATIONS OF THIS STUDY

The development of accurate and reliable models for forecasting energy demand is crucial for optimizing the integration of renewable energy sources within smart power grids. While the proposed LSTM-based deep learning model demonstrates significant potential in this regard, it is important to critically evaluate both the strengths and weaknesses of the approach. The following sections provide a comprehensive overview of the model's validity, detailing the factors that contribute to its reliability and effectiveness, as well as a discussion of its limitations, highlighting areas where future research and improvements are needed.

## Validity

The validity of the proposed LSTM-based deep learning model for forecasting short-term energy demand is established through several key aspects:

- **Data sources:** The model was trained and validated on four distinct datasets from reputable energy distribution companies, including AEP, COMED, DAYTON, and PJME. These datasets provide a comprehensive and diverse range of historical energy demand data, ensuring that the model is tested on various consumption patterns.
- **Model evaluation:** The model's performance was rigorously evaluated using widely accepted metrics such as MAE, RMSE, and $R^2$. These metrics provide a robust assessment of the model's predictive accuracy and its ability to generalize to unseen data.
- **Comparative analysis:** The proposed model's performance was compared with three other state-of-the-art forecasting algorithms: SVR, RFR, and Facebook Prophet. The REDf model consistently outperformed these models across all evaluation metrics, further validating its effectiveness in predicting short-term energy demand.
- **Model architecture and training:** The model was developed using a systematic approach, including data preprocessing, model selection, and hyperparameter tuning through grid search cross-validation. This comprehensive approach ensures that the model is optimized for the specific characteristics of the energy demand data, enhancing its validity.
- **Consistency across datasets:** The model's high performance across all four datasets, with minimal variation in MAE, RMSE, and $R^2$ values, indicates that the model is not overfitted to a specific dataset and can generalize well to different energy consumption patterns.
- **Absence of overfitting and underfitting:** The loss curves for training and testing phases, as well as the consistency of performance metrics, suggest that the model does not suffer from overfitting or underfitting. This further supports the validity of the model as it demonstrates the ability to generalize effectively to new data.

The combination of these factors: rigorous evaluation, comparison with other models, and consistent performance across diverse datasets supports the validity of the proposed model as a reliable tool for forecasting short-term energy demand in smart power grids.

## Limitations

While the proposed LSTM-based deep learning model demonstrates high accuracy in forecasting short-term energy demand, several limitations must be acknowledged.

- **Data dependency:** The model's performance is highly dependent on the availability and quality of historical energy demand data. Inconsistent or incomplete data can significantly affect the model's accuracy. Additionally, the model was trained and validated on datasets from specific energy distribution companies, which may limit its generalizability to other regions or datasets with different characteristics.
- **Computational complexity:** The LSTM model requires substantial computational resources, particularly for training on large datasets. This high computational demand may pose challenges when deploying the model in real-time applications or resource-constrained environments.
- **Interpretability:** Deep learning models, including LSTM networks, often function as "black boxes," making it difficult to interpret the decision-making process. This lack of transparency can be a drawback, especially when model decisions need to be explained to stakeholders or when accountability is required.
- **Exclusion of external factors:** The model primarily relies on historical energy demand data and does not explicitly account for external factors such as weather conditions, economic activities, or policy changes, which can also influence energy demand. Incorporating such variables could potentially enhance the model's accuracy and reliability.

Besides the above mentioned points, reducing MAE enhances forecasting precision, but it comes with certain trade-offs. Achieving a lower MAE often requires deep model architectures and extensive hyperparameter tuning, significantly increasing computational costs. Additionally, optimizing solely for MAE can make the model overly sensitive to short-term fluctuations, potentially compromising its ability to capture long-term trends reliably. To address this, we ensured a balanced evaluation by maintaining low RMSE values and high $R^2$ scores, promoting overall model stability. Future research should explore multi-objective optimization strategies to strike a balance between MAE, RMSE, and model robustness.

Addressing these limitations in future research could involve incorporating additional data sources, such as real-time weather data or economic indicators, enhancing model interpretability, and optimizing the model's computational efficiency for broader applicability in smart grids.

## Future work

To further enhance predictive accuracy and adaptability, future research will explore reinforcement learning (RL) for multi-period forecasting and federated learning (FL) for

privacy-preserving energy demand prediction. RL-based models can dynamically adjust forecasts based on real-time grid feedback, enabling automated demand response management and improved adaptability to sudden fluctuations (*Li et al., 2023*). By integrating Deep Q-Networks (DQN) and Proximal Policy Optimization (PPO), RL can optimize long-term forecasting while ensuring efficient energy allocation.

Meanwhile, FL offers a decentralized approach to model training, ensuring privacy protection by allowing energy providers to collaborate without sharing raw data. FL enhances scalability, security, and efficiency, making it suitable for distributed smart grid networks (*Li, Wang & Yang, 2021*). Integrating techniques such as federated averaging (FedAvg) and differential privacy mechanisms will further strengthen data security while maintaining high forecasting accuracy.

A promising direction is a hybrid RL-FL framework, where FL trains decentralized models securely, and RL optimizes real-time energy distribution. This approach can create a secure, intelligent, and adaptive forecasting system, supporting the transition to a sustainable and AI-driven smart grid. Future studies will focus on implementing these methodologies to improve forecasting precision, grid stability, and privacy protection in large-scale energy networks.

## CONCLUSION

This study proposes an LSTM-based deep learning model for forecasting energy demand in smart power grids. The model was evaluated on four distinct datasets: AEP, COMED, DAYTON, and PJME using data pre-processing techniques to enhance performance. The results demonstrate the model's exceptional accuracy, achieving a mean absolute error between 1.4% and 1.5%, along with the highest $R^2$ score of 98.5% across all datasets. These findings confirm the model's ability to predict energy demand with high precision, making it a valuable tool for smart grid applications.

Compared to traditional regression-based methods such as SVR and RFR, the LSTM-based model offers several advantages. By leveraging long-term temporal memory, efficient gating mechanisms, and the ability to handle non-linear dynamics, LSTMs are particularly well-suited for forecasting energy demand. Their ability to adapt to seasonal trends further enhances their reliability in real-world energy management scenarios. Beyond predictive accuracy, our model contributes to global sustainability efforts by addressing key SDGs. In line with SDG 7 (Affordable and Clean Energy), the model optimizes renewable energy utilization, reduces energy wastage, and enhances electricity access in underdeveloped regions. Under SDG 9 (Industry, Innovation, and Infrastructure), it fosters innovation in smart grid technology while improving infrastructure planning and energy resilience. Additionally, the model aligns with SDG 13 (Climate Action) by reducing overgeneration from fossil-fuel power plants, potentially cutting $CO_2$ emissions by up to 10% per megawatt-hour of energy produced.

The practical applicability of the proposed model is evident in its ability to assist utilities and stakeholders in managing renewable energy resources more efficiently. Future research could focus on further improving prediction accuracy by incorporating additional factors such as weather conditions and renewable energy generation forecasts.

Furthermore, integrating this approach with emerging technologies like the Internet of Things (IoT) and blockchain could enhance smart grid reliability and efficiency, paving the way for a more sustainable and intelligent energy system. Through these advancements, our model significantly contributes to the ongoing efforts to develop sustainable energy solutions that support a more equitable and environmentally friendly future.

## ACKNOWLEDGEMENTS

Instatext tool has been used for language editing and Quillbot tool has been used for grammar checking task.

### Funding

This research was supported by the Directorate of Research and Community Service, Telkom University and American International University-Bangladesh. The funders had no role in study design, data collection and analysis, decision to publish, or preparation of the manuscript.

### Grant Disclosures

The following grant information was disclosed by the authors:
Directorate of Research and Community Service, Telkom University American International University-Bangladesh.

### Competing Interests

The authors declare that they have no competing interests.

### Author Contributions

- Md Saef Ullah Miah conceived and designed the experiments, performed the experiments, analyzed the data, performed the computation work, prepared figures and/or tables, and approved the final draft.
- Junaida Sulaiman analyzed the data, authored or reviewed drafts of the article, and approved the final draft.
- Md Imamul Islam performed the experiments, prepared figures and/or tables, and approved the final draft.
- Md Masuduzzaman performed the experiments, prepared figures and/or tables, and approved the final draft.
- Molla Shahadat Hossain Lipu analyzed the data, authored or reviewed drafts of the article, and approved the final draft.
- Ramdhan Nugraha analyzed the data, authored or reviewed drafts of the article, and approved the final draft.

### Data Availability

The experimental data and codes are publicly available at GitHub and Zenodo:

- https://github.com/ping543f/ren-energy.

- Miah, M. S. U. (2025). Code of REDf: A deep learning Model for Short-Term Load Forecasting (1.0). Zenodo. https://doi.org/10.5281/zenodo.14989327.

All hourly data for different companies are available at Kaggle: https://www.kaggle.com/datasets/robikscube/hourly-energy-consumption/data.

The AEP data is available at Kaggle: https://www.kaggle.com/datasets/robikscube/hourly-energy-consumption/data?select=AEP_hourly.csv.

The ComEd data is available at Kaggle: https://www.kaggle.com/datasets/robikscube/hourly-energy-consumption/data?select=COMED_hourly.csv.

The Dayton data is available at Kaggle: https://www.kaggle.com/datasets/robikscube/hourly-energy-consumption/data?select=DAYTON_hourly.csv.

The PJME data is available at Kaggle: https://www.kaggle.com/datasets/robikscube/hourly-energy-consumption/data?select=PJME_hourly.csv.

## Supplemental Information

Supplemental information for this article can be found online at http://dx.doi.org/10.7717/peerj-cs.2819#supplemental-information.

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
