# Peer review of "REDf: a deep learning model for short-term load forecasting to facilitate renewable integration and attaining the SDGs 7, 9, and 13"

_PeerJ Computer Science, doi:10.7717/peerj-cs.2819_

## Round 0.1 · original submission · Major Revisions

In the paper, the LSTM-based energy demand forecasting model presents interesting findings, but lacks innovation and justification. In particular, it highlights that the methodology is not sufficiently compared with existing studies in the literature and that there are shortcomings in the methodology (data processing details, lack of more up-to-date comparative models).

Reviewer 1 ·

Basic reporting

This research introduces a sophisticated deep learning model leveraging LSTM networks to enhance the accuracy of short-term energy demand predictions in smart grids. In general, this work presents some interesting findings. However, it has several concerns that need to be alleviated.

1 Considering the inherent variability of renewable energy sources, how does your model maintain high predictive accuracy during periods of peak fluctuations in renewable generation? Please elaborate on any mechanisms implemented to stabilize predictions under such volatile conditions.

2 Among the features of LSTM networks, which do you consider most critical for achieving superior predictive performance in smart grid applications compared to traditional models like SVR or Random Forest? Could you detail the technical aspects that give LSTM an advantage in this context?

3 In alignment with Sustainable Development Goals 7, 9, and 13, could you specify the quantifiable impacts of your model on advancing sustainable and innovative energy infrastructures? How does your research contribute to the broader agenda of ensuring sustainable energy access and enhancing infrastructure resilience?

4 Overfitting presents significant challenges in predictive modeling, especially when integrating diverse data types and managing large datasets. How have you addressed this issue in your LSTM model, and what strategies have you found most effective in maintaining model generalizability?

5 What methodologies did you employ to validate the predictive accuracy of your model against real-world energy consumption data, and what was the consistency of your model’s performance across various datasets? Please provide details on any statistical tests or benchmarks used in the validation process.

6 Your approach to minimizing Mean Absolute Error (MAE) is crucial for operational efficiency in smart grids. Could you discuss the trade-offs involved in focusing on MAE reduction? How does this focus affect other performance metrics and the practical usability of the model in operational environments?

7 Given the recent advancements in automated reinforcement learning for multi-period forecasting and federated deep reinforcement learning for data privacy, this reviewer recommends discussing how these methodologies could enhance the predictive accuracy and privacy capabilities of your LSTM model. Integrating relevant and contemporary academic sources, such as the studies at DOI: 10.1016/j.apenergy.2022.120291 and DOI: 10.1109/TSTE.2021.3105529, will substantiate your paper’s claims and enrich the context for your readers

Experimental design

no comment

Validity of the findings

no comment

Additional comments

no comment

·

Basic reporting

The paper is written & structured well. The technical scope of the work covers the critical aspect of data forces acting using advanced artificial intelligence methods.

Minor amendment recommended at Literature review section :
Group the articles reviewed under some common criteria; Therefore, the literature review needs sub-sections.

Experimental design

no comment

Validity of the findings

The quality of Figures 6 to 9 needs major improvements for the actual Vs predicated data sets for the following:
Figure 6. Actual vs. Predicted data for different models for AEP dataset
Figure 7. Actual vs. Predicted data for different models for COMED dataset
Figure 8. Actual vs. Predicted data for different models for DAYTON dataset.
Figure 9. Actual vs. Predicted data for different models for PJME dataset.

Additional comments

Additional graphs are needed to compare all four datasets in one graph.

Reviewer 3 ·

Basic reporting

Good

Experimental design

Good

Validity of the findings

Good

Additional comments

as above

Reviewer 4 ·

Basic reporting

- The manuscript introduces an LSTM-based model for forecasting energy demand; however, this approach has been extensively explored in existing literature. The proposed method, named "REDf," does not clearly establish its novelty compared to prior studies.
- Although LSTMs are appropriate for time series data, the manuscript does not adequately justify why this model was chosen over more advanced methods, such as Transformer models, which have shown superior performance in sequential data tasks.
- The discussion on aligning the study with SDGs 7, 9, and 13 is superficial. While the manuscript claims contributions to these goals, it fails to provide concrete or measurable evidence to support these assertions.

Experimental design

- The selection of baseline models (SVR, RFR, Facebook Prophet) is outdated and does not adequately demonstrate the model's superiority. Including more recent or hybrid approaches would provide a more comprehensive comparison.
- The datasets employed (AEP, COMED, DAYTON, PJME) are restricted to the U.S., raising concerns about the model’s applicability to other regions with different energy consumption patterns.
- While the manuscript mentions data normalization and outlier handling, it lacks detailed explanations or visual representations of these pre-processing steps, raising doubts about reproducibility and their impact on the results.
- The evaluation metrics used (MAE, RMSE, R²) are standard for forecasting but insufficient to fully evaluate the model’s performance. Additional metrics like MAPE or energy-specific reliability indicators would provide more comprehensive insights.

Validity of the findings

- The manuscript asserts that the model avoids overfitting but does not present robust evidence, such as cross-validation results or detailed plots showing training and validation losses over multiple iterations.
- Although figures illustrating actual versus predicted data are included, their clarity and consistency need improvement. Providing more detailed and properly labeled visualizations would enhance understanding of the model’s performance and limitations.

Additional comments

- The discussion section lacks specific and actionable suggestions for future research. Including recommendations such as incorporating external factors (e.g., weather patterns or policy impacts) or testing the model on real-time streaming data would add value to the study

---

## Round 0.2 · accepted · Accept

The revisions have been completed successfully and the improvements have significantly increased the contribution of the study. Accordingly, I recommend that the article be accepted for publication.

Reviewer 1 ·

Basic reporting

The manuscript is clearly written with professional English, and the introduction and background sections have been significantly improved, providing sufficient context and motivation.

Relevant literature is comprehensively referenced, aligning the manuscript with current trends and standards in deep learning-based load forecasting.

Experimental design

The revised manuscript presents rigorous experimental procedures. Data preprocessing steps, model implementation, and hyperparameter tuning via grid search are now clearly described, enhancing reproducibility.

The authors have also provided adequate justification for their model choice (LSTM) and comparative models (SVR, Random Forest, and Prophet), with well-articulated reasons for model selection.

Validity of the findings

The results section now clearly demonstrates the proposed REDf model’s superior performance across multiple metrics (MAE, RMSE, R², and MAPE) and datasets. This strongly supports the model's validity and applicability in short-term load forecasting.

Limitations and future research directions are now explicitly stated, strengthening the manuscript's conclusions.

Additional comments

Overall, the revised manuscript demonstrates significant improvement, clearly addressing previous comments and enhancing the quality and clarity of the research. I find this version acceptable for publication.

Reviewer 3 ·

Basic reporting

Accept no more comments from my side

Experimental design

good

Validity of the findings

good

Additional comments

no

Reviewer 4 ·

Basic reporting

Authors has addressed my comments. The revised paper is ready to go.

Experimental design

Authors has addressed my comments. The revised paper is ready to go.

Validity of the findings

Authors has addressed my comments. The revised paper is ready to go.

Additional comments

Authors has addressed my comments. The revised paper is ready to go.